# The nuclear receptor ERR cooperates with the cardiogenic factor GATA4 to orchestrate cardiomyocyte maturation

Tomoya Sakamoto [1], Kirill Batmanov [2], Shibiao Wan [2,4], Yuanjun Guo [1,5], Ling Lai[1], Rick B. Vega[3,6] & Daniel P. Kelly [1✉]

Estrogen-related receptors (ERR) α and γ were shown recently to serve as regulators of cardiac maturation, yet the underlying mechanisms have not been delineated. Herein, we find that ERR signaling is necessary for induction of genes involved in mitochondrial and cardiac-specific contractile processes during human induced pluripotent stem cell-derived cardio-myocyte (hiPSC-CM) differentiation. Genomic interrogation studies demonstrate that ERRγ occupies many cardiomyocyte enhancers/super-enhancers, often co-localizing with the cardiogenic factor GATA4. ERRγ interacts with GATA4 to cooperatively activate transcription of targets involved in cardiomyocyte-specific processes such as contractile function, whereas ERRγ-mediated control of metabolic genes occurs independent of GATA4. Both mechanisms require the transcriptional coregulator PGC-1α. A disease-causing GATA4 mutation is shown to diminish PGC-1α/ERR/GATA4 cooperativity and expression of ERR target genes are downregulated in human heart failure samples suggesting that dysregulation of this circuitry may contribute to congenital and acquired forms of heart failure.

[1] Cardiovascular Institute, Department of Medicine, Perelman School of Medicine at the University of Pennsylvania, Philadelphia, PA 19104, USA. [2] Institute for Diabetes, Obesity and Metabolism, Department of Medicine, Perelman School of Medicine at the University of Pennsylvania, Philadelphia, PA 19104, USA. [3] Center for Metabolic Origins of Disease, Sanford Burnham Prebys Medical Discovery Institute, Orlando, FL 32827, USA. [4] Present address: Center for Applied Bioinformatics, St. Jude Children's Research Hospital, Memphis, TN 38105, USA. [5] Present address: Biomarker Discovery, Amgen Research, Amgen Inc., 1120 Veterans Blvd, South San Francisco, CA 94080, USA. [6] Present address: Research and Early Development, Cardiovascular, Renal, and Metabolism (CVRM), BioPharmaceuticals R&D, AstraZeneca, Gaithersburg, MD 20878, USA. ✉email: dankelly@pennmedicine.upenn.edu

The heart requires an extraordinary capacity for mitochondrial ATP production to match the high energy demands of a persistent pump. The normal adult heart relies heavily on fatty acids as a fuel using a specialized high-capacity mitochondrial fatty acid oxidation (FAO) system[1]. Cardiac postnatal development begins with a dramatic mitochondrial biogenic response at birth, followed by a mitochondrial maturation phase in which the capacity for FAO and respiration are markedly increased due, in large part, to transcriptional induction of genes involved in these energy transduction pathways. This maturation process also involves a shift from fetal to adult isoforms of many cardiac-specific structural genes involved in ATP utilization processes such as contractile function and ion transport. The inducible transcriptional coactivators peroxisome proliferator-activated receptor gamma coactivator 1 (PGC-1) α and β are necessary for the perinatal cardiac mitochondrial expansion[2]. PGC-1α serves as a coactivator of the nuclear receptors, peroxisome proliferator-activated receptor (PPAR) α[3] and estrogen-related receptors (ERRs)[4], both of which direct the transcription of genes involved in postnatal mitochondrial maturation. Notably, during the development of cardiac hypertrophy, the PGC-1/PPAR/ERR circuit becomes partially deactivated resulting in the re-expression of fetal energy metabolic programs that likely contribute to the pathogenesis of heart failure[5–7].

Using conditional gene targeting strategies during postnatal development, we and others have shown that ERRα and γ, key downstream effectors of PGC-1α[8,9], are necessary for normal mitochondrial maturation of the mouse heart[10,11]. Surprisingly, we found that in addition to its role in mitochondrial function, ERR signaling is also necessary for a broader postnatal cardiac maturation program including induction of adult contractile and ion transport genes[10]. In addition, ERR deficiency resulted in the abnormal re-expression of many early developmental and non-cardiac lineage genes in the developing mouse heart, suggesting that it serves to repress these genes during cardiac myocyte differentiation[10]. Cistromic analyses in human induced pluripotent stem cell-derived cardiomyocytes (hiPSC-CMs) suggested that ERRγ is a direct activator of adult contractile and ion channel genes in addition to regulating canonical target genes involved in mitochondrial energetic processes such as FAO and oxidative phosphorylation (OXPHOS)[4,7,10].

The observation that ERR signaling serves a critical role in the transcriptional control of a broad array of cardiac postnatal developmental processes raises the broader question of how genes involved in mitochondrial function, a ubiquitous and fundamental function, are coordinately regulated with cell-specific energy-consuming processes such as cardiac contractile function? The answer to this question will provide insight into how the capacity for cellular ATP production is matched to energy demands under normal conditions, and how it may become disrupted in disease states such as heart failure.

Herein we describe a mechanism whereby ERR signaling coordinates transcriptional control of diverse transcriptional programs involved in cardiomyocyte differentiation. Using hiPSC-CMs as a model system, genomic interrogation demonstrated that ERRγ serves as a key component of cardiac myocyte enhancers and super-enhancers to control both energy metabolic and cardiac structural genes. We found that ERR-mediated regulation of cardiac-specific genes, such as those involved in postnatal contraction and ion transport, involves a cooperative interaction with the cardiogenic factor GATA-binding protein 4 (GATA4). In contrast, the regulation of canonical energy metabolic gene targets by ERR occurs largely independent of GATA4. The transcriptional regulator PGC-1α serves to coactivate both types of processes providing a common integrative pathway.

Lastly, we found that a known human cardiomyopathy-causing GATA4 mutant reduces PGC-1/ERRγ/GATA4 cooperativity and that many GATA-dependent and GATA4-independent ERR target genes are downregulated in ventricular samples from humans with heart failure.

## Results

**ERRγ directs cardiac maturation programs.** hiPSC-CMs were employed as a model system to investigate the role of ERR signaling during cardiomyocyte differentiation. Transcripts encoding the ERR family of nuclear receptors and the upstream coactivator PGC-1α (Fig. 1a), as well as protein levels of ERRγ (Fig. 1b) were markedly induced during hiPSC-CM differentiation in parallel with expression of genes encoding cardiogenic factors GATA4 and MEF2C (Fig. 1a) and sarcomere α-Actinin protein (Fig. 1b). Notably, expression of mitochondrial genes (COX6A2, PDK4, MDH1, and CKMT2) and a key adult sarcomere gene marker (TNNI3) were coordinately induced during hiPSC-CM differentiation (Supplementary Fig. 1a).

hiPSC-CMs lacking ERRγ (ERRγ KO) exhibited significant downregulation of known ERRγ target genes encoding mitochondrial and adult contractile markers on day 22 of cardiac differentiation (representative genes in Supplementary Fig. 1b). Given that ERRγ and ERRα regulate overlapping targets in cardiac myocytes[10–12], we next employed hiPSC-CMs lacking both ERRα and ERRγ[10]. RNA-sequencing (RNA-seq) analysis was conducted on two independent ERRα/γ KO hiPSC-CM lines (KO1 and KO6). Compared to wild-type (WT) control hiPSC-CMs, there were 4,038 differentially expressed (DE) genes (2226 upregulated, 1812 downregulated) in line KO1, and 4913 DE genes (2535 upregulated, 2378 downregulated) in line KO6 (|fold change (FC)| > 1.5, Benjamini–Hochberg false discovery rate (FDR) < 0.05; Supplementary Fig. 1c). Enrichment analysis of DE genes in both lines of ERRα/γ KO hiPSC-CMs demonstrated downregulation of a wide array of genes involved in mitochondrial metabolic and cardiac-specific structural processes (Supplementary Fig. 1d). Specifically, expression of genes involved in the adult cardiac sarcomere machinery (e.g MYH7, MYBPC3, TNNC1, MYL2, and TNNI3), Ca²⁺ handling (e,g, ATP2A2, PLN, and RYR2), and ion transport (ATP1A3, KCNJ8, and KCNQ1, etc.) were downregulated in ERRα/γ KO hiPSC-CMs (Fig. 1c). Notably, ERRα/γ-deficiency impacted expression of a subset of well-established adult cardiac sarcomeric gene isoforms as demonstrated by marked reduction in expression of TNNI3, MYH7, and MYL2 with only minimal impact on the fetal form, TNNI1 (Fig. 1c; Supplementary Fig. 1e). The protein levels of MYL2, MYBPC3, and TNNI3 were also downregulated by loss of ERRα/γ in hiPSC-CMs without changing cardiac specification as evidenced of GATA4 protein expression (Fig. 1d), and the number of cardiac Troponin T-positive cells[10] suggesting ERRα/γ is an essential activator for genes coding cardiac adult structural programs.

The impact of ERRγ overexpression (OE) was also assessed using adenovirus expressing FLAG-tagged ERRγ (Ad-ERRγ, Supplementary Fig. 1f). RNA-seq analysis of ERRγ OE hiPSC-CMs revealed 2540 upregulated and 2350 downregulated genes 48 h after the infection of Ad-ERRγ compared to Ad-GFP (|FC| > 1.5, Benjamini–Hochberg FDR < 0.05; Supplementary Fig. 1g). The enrichment analysis and its qPCR validation demonstrated that ERRγ OE resulted largely in the induction of genes involved in mitochondrial oxidative metabolism including FAO, OXPHOS, the tricarboxylic acid (TCA) cycle, and branched-chain amino acid (BCAA) catabolism (Fig. 1e, Supplementary Figs. 1h and 2a, b). In addition, expression of the adult contractile maturation marker TNNI3 was moderately but

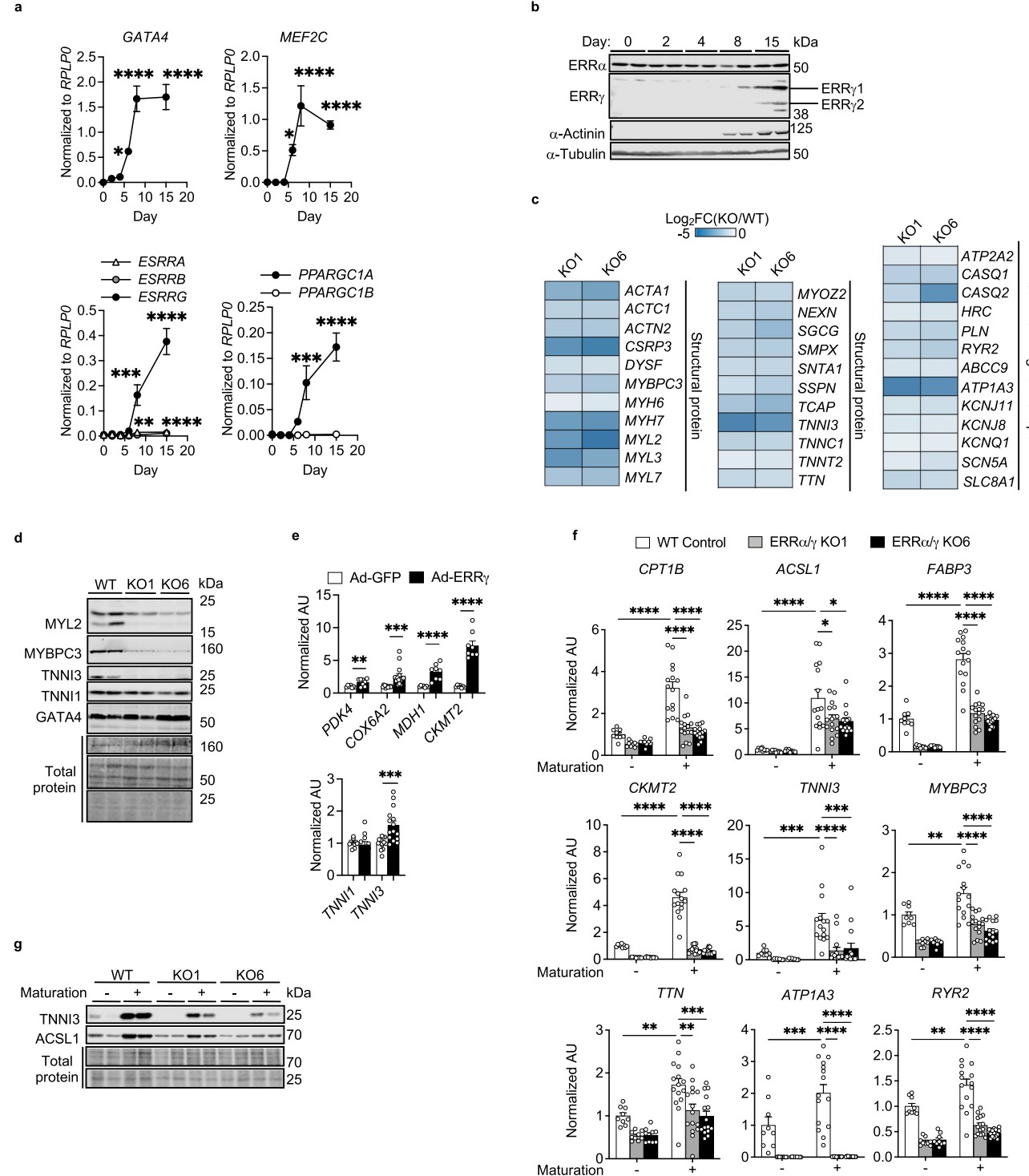

significantly upregulated, while *TNNI1* expression was unchanged by ERRγ OE (Fig. 1e).

A key feature of cardiac myocyte differentiation is development of high-capacity mitochondrial FAO, the main source of ATP production in the adult heart. *ACSL1* encodes long-chain acyl-CoA synthetase, a mitochondrial enzyme that catalyzes the thio-esterification of long-chain fatty acids prior to entering mitochondrial β-oxidation[13]. Both mRNA and protein levels of ACSL1, the adult isoform, but not ACSL3 (fetal isoform) were markedly induced in ERRγ OE hiPSC-CMs (Supplementary Fig. 2a) resulting in a marked increase in ACSL1/3 ratio

(Supplementary Fig. 2b) as occurs during the fetal to adult transition in vivo[10,14] Consistent with the gene expression profile, ERRγ OE increased $^3$H-palmitate oxidation rates (Supplementary Fig. 2c) together with an increase in acylcarnitine species indicative of increased FAO flux (Supplementary Fig. 2d). Other metabolite signatures of metabolic maturation in ERRγ OE hiPSC-CMs included increased levels of TCA cycle intermediates (Supplementary Fig. 2e), reduced lactate consistent with decreased glycolytic flux (Supplementary Fig. 2e), and increased levels of mitochondrial BCAA degradation products (C3, C4, and C5 acylcarnitine species; Supplementary Fig. 2f). Levels of

**Fig. 1 ERR signaling is necessary and sufficient for energy metabolic and structural maturation programs in hiPSC-CMs. a** Real-time quantitative polymerase chain reaction (RT-qPCR)-determined levels of the designated mRNAs during human induced pluripotent stem cell-derived cardiomyocyte (hiPSC-CM) differentiation. Levels of indicated genes are shown normalized to *RPLP0* levels at each timepoint. Day 0 and 2, $n = 9$; Day 4, 6, 8, and 15, $n = 8$. *$p < 0.05$, ***$p < 0.001$, ****$p < 0.0001$ vs Day 0; one-way ANOVA followed by Dunnett's multiple comparison test. **b** Representative immunoblots of estrogen-related receptor (ERR) α and γ protein levels during hiPSC-CM differentiation. α-Tubulin was used as a loading control. **c** Heatmap representing $\log_2$ fold change [ERRα/γ knockout (KO)/wild-type control (WT)] of mRNA levels of genes involved in cardiac structural component, $Ca^{2+}$ handling function, and ion transport in two different KO cell lines (KO1, KO6). All changes were significant (Benjamini-Hochberg false discovery rate <0.05) except for *SCN5A* in KO1 ($p = 0.057$). **d** Representative immunoblots of indicated protein levels in WT Control and ERRα/γ KO hiPSC-CMs. Total protein staining was shown as a loading control. **e** The mRNA levels of indicated genes determined by RT-qPCR in hiPSC-CMs following the infection of adenovirus expressing GFP or ERRγ (Ad-GFP or Ad-ERRγ). **$p < 0.01$, ***$p < 0.001$, ****$p < 0.0001$, Ad-ERRγ vs Ad-GFP, two-tailed student's t-test. *PDK4* (Ad-GFP, $n = 7$ and Ad-ERRγ, $n = 8$); *COX6A2* (Ad-GFP, $n = 14$ and Ad-ERRγ, $n = 15$); *MDH1* and *CKMT2*, (Ad-GFP, $n = 7$ and Ad-ERRγ, $n = 8$); *TNNI3* (Ad-GFP, $n = 14$ and Ad-ERRγ, $n = 15$); *TNNI1* (Ad-GFP, $n = 10$ and Ad-ERRγ, $n = 11$). **f** The mRNA levels of indicated hiPSC-CM maturation markers determined by RT-qPCR in WT and ERRα/γ KO hiPSC-CMs derived from KO1 and KO6 with ($n = 15$) or without ($n = 9$) maturation cocktail treatment for 7 days. *$p < 0.05$, **$p < 0.01$, ***$p < 0.001$, ****$p < 0.0001$, WT vs ERRα/γ KO hiPSC-CMs, two-way ANOVA followed by Tukey's multiple comparisons test. All graphs in **a**, **e**, and **f** represent the means ± SEM. *n* denotes independent biological replicates. **g** Representative immunoblot images to show ACSL1 and TNNI3 protein expression levels in WT and ERRα/γ KO hiPSC-CMs with or without maturation cocktail. Total protein staining was used as a loading control.

transcript encoding BCAA enzymes were also increased in ERRγ OE hiPSC-CMs (Supplementary Fig. 2g).

Recently, a protocol to drive stem cell-derived CM to a more mature state in culture was described[15] using culture media supplemented with palmitate and several hormonal agonists including triiodothyronine, GW7647:PPARα agonist, and dexamethasone. We found that addition of this supplemented media to our differentiation protocol significantly increased levels of maturation genes including those involved in FAO (*FABP3*, *ACSL1*, *CPT1B*, *CKMT2*), sarcomeric function (*TNNI3*, *MYBPC3*, *TTN*), ion transport (*ATP1A3*), and $Ca^{2+}$ handling (*RYR2*) as shown in Fig. 1f. The protein levels of structural and metabolic maturation markers such as TNNI3 and ACSL1 were also highly induced with the defined media (Fig. 1g). The expression of this panel of maturation markers was markedly suppressed in ERRα/γ KO hiPSC-CMs (Fig. 1f–g). These results further support the conclusion that ERR signaling functions as a driver of hiPSC-CM maturation.

**ERRγ serves a key role in cardiomyocyte enhancers.** Intersection analysis of an ERRγ chromatin immunoprecipitation sequencing (ChIP-seq) dataset [10; GSE113784] with the downregulated genes in the ERRα/γ KO hiPSC-CM RNA-seq dataset (Supplementary Fig. 3a) demonstrated that ERRγ directly activates transcription of cardiac myocyte genes involved in diverse maturation processes including mitochondrial fuel and energy metabolism and contractile function (Supplementary Fig. 3b). We next sought to determine the potential role of ERRγ in cardiac myocyte enhancers. To this end, ChIP-seq studies were conducted with hiPSC-CMs using antibodies directed against acetylated histone 3 at lysine residue 27 (H3K27ac, Supplementary Fig. 3c), an active enhancer mark[16,17], and compared with the ERRγ cistrome. 22.8% of ERRγ peaks (7567/33,253 peaks) overlapped with H3K27ac peaks, and, conversely, 26.5% of H3K27ac-occupied regions (7567/28,602 peaks) had ERRγ peaks in WT hiPSC-CMs (Fig. 2a). H3K27ac deposition around ERRγ peaks was significantly reduced by ERRγ KO (Fig. 2b). As a control, we plotted H3K27ac ChIP-seq signals around GATA6, another cardiac transcription factor, in hiPSC-CMs[18]. H3K27ac deposition was higher around ERRγ than around GATA6, and the decreased H3K27ac signals by KO ERRγ occurred specifically around ERRγ peaks (Fig. 2b).

Motif analysis of the peaks that exhibited decreased H3K27ac deposition showed enrichment of myocyte enhancer factor-2 (MEF2), GATA, and transcriptional enhanced associate domain (TEAD) binding sites in addition to the ERR motif (Fig. 2c). We

observed a similar trend in the enriched motifs on ERRγ peaks with H3K27ac signals that were not altered by KO ERRγ (Supplementary Fig. 3d). Enrichment analysis of genes associated with ERRγ peaks that demonstrated decreased H3K27ac deposition in the ERRγ KO hiPSC-CMs identified targets involved in both mitochondrial energy metabolism (e.g. *CPT1B* and *COX6A2*) and cardiac contractile function (*TNNI3* and *MYL2*; Fig. 2d–e). Notably, ERR-activated gene targets (downregulated genes in ERRα/γ KO hiPSC-CMs) generally exhibited decreased H3K27ac deposition by KO ERRγ, whereas the majority of ERR-suppressed genes did not exhibit alterations in H3K27ac deposition, and the H3K27ac peaks were generally smaller (Fig. 2f–g). NR5A2, thyroid hormone receptor, and MEF2 motifs as well as the ERR binding motif were enriched at the location of ERRγ peaks (Supplementary Fig. 3e).

We next sought to determine whether ERRγ served as a component of cardiac super-enhancer (SE) regions. For these analyses, we intersected our hiPSC-CM ERRγ ChIP-seq dataset with the results of a published Mediator complex 1 (MED1) ChIP-seq analysis conducted in hiPSC-CMs that identified 213 SE sites (GSE85631)[19] (Fig. 3a). Interestingly, fifty-seven percent of the SEs identified previously (122 sites) were found to overlap ERRγ peaks (SE + ERRγ; Fig. 3a and Supplementary Data 1). In marked contrast to the ERR enhancer results, the vast majority of the ERRγ-containing SEs overlapped with proximal promoter regions linked to twenty cardiac-enriched structural genes including *TNNT2*, *MYL3*, *ACTN2*, *TTN* and but only a very small number of cardiac-enriched metabolic genes including *FABP3*, *GPAT3*, *PGM1*, *and PYGB* (full gene list in Supplementary Table 1, representative genomic browsers shown in Fig. 3b). In addition, recently described evolutionarily conserved cardiac SEs associated with the *NPPA-NPPB*[20] and *MYH6-7* cluster[21] overlapped ERRγ peaks (Fig. 3b). Levels of transcripts encoded by 73.1% of annotated genes near ERRγ-containing SEs were reduced by ERRα/γ deficiency (68/93 genes; Fig. 3c, d and Supplementary Table 1). Taken together, these results indicate that ERRγ occupies a significant subset of cardiac enhancer/ promoter and SE regions with the latter largely regulating cardiac-specific genes.

The role of ERRs on chromatin accessibility was next assessed using Assay for Transposase Accessible Chromatin with high-throughput sequencing (ATAC-seq) in hiPSC-CMs (Supplementary Fig. 4). 17,588 open chromatin sites were defined by ATAC-seq, 68.1% of which contained H3K27ac, indicative of active enhancer or promoter regions (Fig. 4a). 18.3% of highly accessible regions contained ERRγ peaks (3218/17,588). In addition, 73.2%

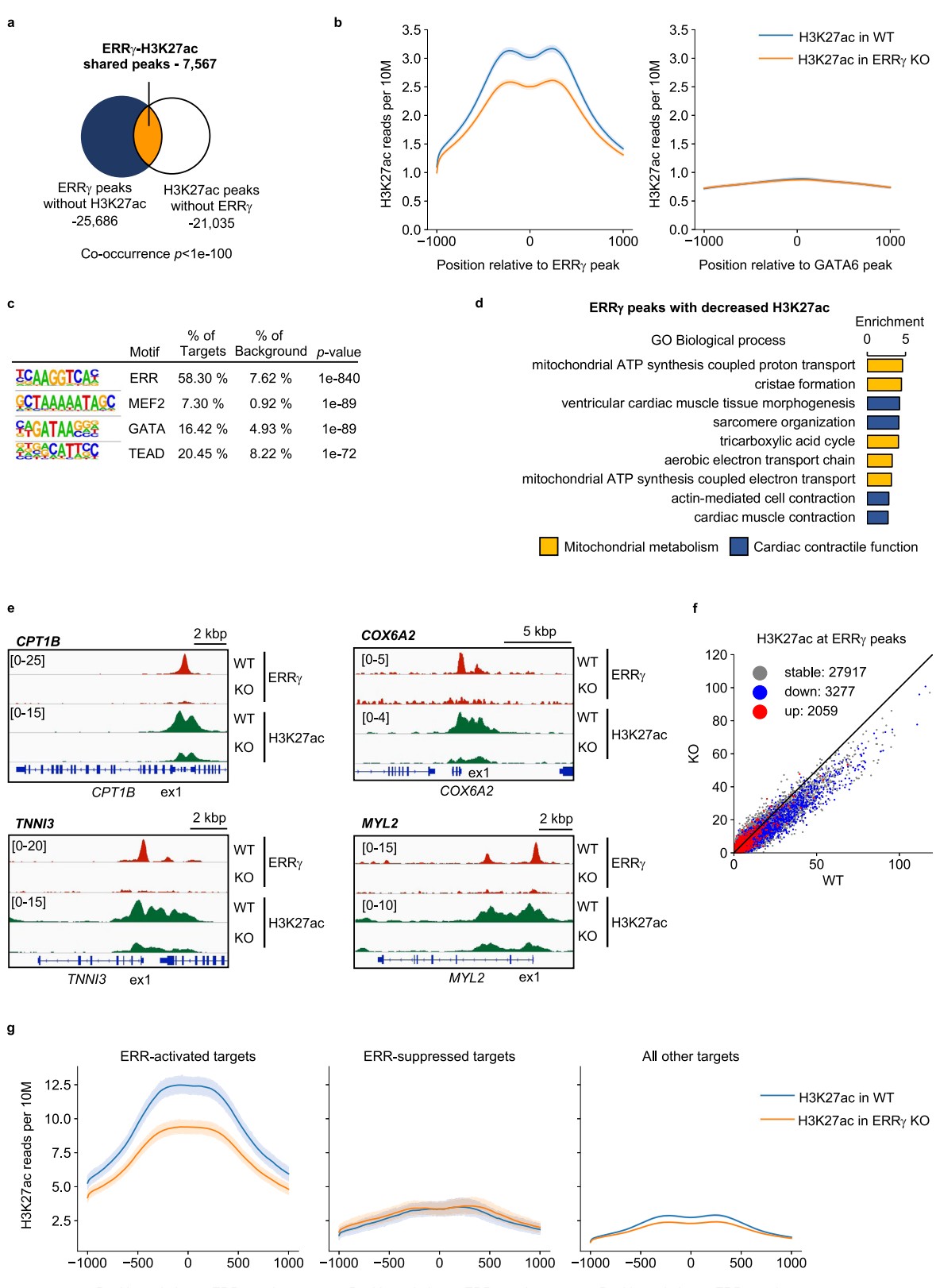

of published SEs[19] (156/213 sites) overlapped with the open chromatin sites defined by the ATAC-seq in hiPSC-CMs (Fig. 4b). The overlapped sites were largely associated with cardiac-enriched genes such as *TNNT2*, *TTN*, *ACTN2*, *PLN*, and *CSRP3*. We observed higher MED1 and H3K27ac depositions in highly accessible regions that contained ERRγ occupation compared to the regions lacking ERRγ (Fig. 4c). The impact of ERR deficiency was assessed by conducting ATAC-seq in ERRα/γ KO hiPSC-CM. ERRα/γ deficiency resulted in significant alterations in the ATAC-seq profile including regions with both reduced (4510 peaks) and increased (2316 peaks) peak sizes (Benjamini–Hochberg FDR < 0.05, |FC| > 2, Fig. 4d). 32.6% (1470/4510 peaks) of decreased

**Fig. 2 ERRγ regulates cardiac enhancer activity in hiPSC-CMs. a** Venn diagram indicates the significantly overlapped peaks (Fisher's exact test, orange color) from ERRγ chromatin immunoprecipitation sequencing (ChIP-seq; GSE113784) and H3K27ac ChIP-seq (GSE165965) in human induced pluripotent stem cell-derived cardiomyocytes (hiPSC-CMs). **b** Aggregation plots represent H3K27ac depositions around total ERRγ or GATA6 peaks (±1kbp) in wild-type control (WT) and ERRγ knockout (KO) hiPSC-CMs. **c** De novo motif enrichment of ERRγ peaks with decreased H3K27ac signals by KO ERRγ. *p*-values were calculated with Fisher's exact test. **d** Bar graphs represent enrichment score of Gene Ontology (GO) Biological Process using the genes associated with ERRγ peaks with decreased H3K27ac depositions by KO ERRγ. Yellow bars indicate the terms related to mitochondrial metabolism and blue bars indicate the terms related to cardiac contractile function. **e** Representative genomic browser tracks of loci coding oxidative metabolic genes and cardiac structural genes. Red peaks represent ERRγ peaks and green peaks represent H3K27ac depositions in WT and ERRγ KO hiPSC-CMs. **f** Scatter plot of ERRγ peaks plotted with H3K27ac ChIP-seq signals. The color indicates the expression changes in ERRα/γ KO hiPSC-CMs. Blue, red, and gray dots represent the downregulated genes, upregulated genes, and not-regulated (stable) genes in ERRα/γ KO hiPSC-CMs respectively. **g** Aggregation plots of H3K27ac depositions around ERRγ peaks on down (ERR-activated targets), upregulated genes (ERR-suppressed targets) in ERRα/γ KO hiPSC-CMs or all other ERRγ peaks (TSS ± 5 kbp). The mean tag count and 95% confidence interval are plotted for each position.

ATAC signals and 7.8% (181/2316 peaks) of increased ATAC signals overlapped documented ERRγ peaks in hiPSC-CMs, suggesting ERR might be involved in the regulation of cardiac chromatin accessibility (Fig. 4d). The ATAC peaks that were decreased by ERRα/γ KO were mainly located in intergenic and intron regions (Fig. 4e), which are typical locations for enhancers[22,23]. In contrast, ATAC peaks that increased with ERRα/γ deficiency were often located in promoter regions (−300 to +50 bp from gene TSS) (Fig. 4e). Aggregation plots show that overall chromatin accessibility around ERRγ peaks (±1kbp) were significantly attenuated by ERRα/γ KO, although this trend was not observed around GATA6 peaks[18] (Fig. 4f). Specifically, 46% (1470/3218 peaks) of ERRγ-associated accessible regions were altered in ERRα/γ KO hiPSC-CMs. As an example, the ATAC-seq signal for *MYL3*, an adult ventricular contractile protein gene locus, colocalized with ERRγ, H3K27ac, and MED1 peaks and the ATAC-seq signals were attenuated by ERRα/γ KO (Fig. 4g). The ERR binding motif was significantly enriched in the ATAC-seq peaks that were reduced by ERRα/γ KO along with sites for known cardiac-enriched transcription factors including MEF2, GATA, heart and neural crest derivatives expressed 2 (HAND2), T-box transcription factor 20 (TBX20), and TEAD (Fig. 4h). Notably, binding motifs for CTCF and BORIS, well-known transcription insulators/repressors, were enriched in the regions that exhibited increased accessibility with ERRα/γ KO (Fig. 4h) suggesting that ERR functions as a repressor to suppress expression of some non-cardiac lineage genes. Taken together, these data demonstrate that ERRα/γ not only functions in enhancer and super-enhancer regions but also modulates chromatin accessibility on a subset of cardiac myocyte genes.

The results of the ATAC-seq analysis suggested that in addition to direct regulation, ERR may influence chromatin accessibility through indirect mechanisms given that a significant number of ATAC-seq peaks altered by ERRα/γ deficiency were not associated with ERR targets. One potential mechanism for an indirect effect could be through regulation of chromatin modifiers by ERRs. Indeed, we found that the expression of the gene encoding the cardiac-enriched histone modifier SET and MYND domain containing 1 (SMYD1), a histone-lysine N-methyltransferase, is diminished by ERRα/γ deficiency (Fig. 3c, d, and Supplementary Fig. 5a–c). In addition, a luciferase reporter of *SMYD1* promoter region which contains ERRγ binding sites (Supplemental Fig. 5b; 24 putative ERR binding sites were predicted by JASPAR[24,25] in this region) was activated by ERRα or ERRγ in the presence of their coactivator PGC-1α (Supplementary Fig. 5c). These results suggest that ERR may also activate cardiac chromatin accessibility indirectly through regulation of SMYD1 and perhaps other histone-modifying enzymes.

**ERRγ activates cardiac gene transcription with GATA4.** The observed ERR-mediated transcriptional control of canonical metabolic genes as well as cardiac-enriched structural genes suggested independent regulatory mechanisms that may involve distinct transcriptional complexes. As a first step to address this possibility, we analyzed the DNA-binding motifs in ERRγ binding regions defined by the ERRγ ChIP-seq analysis. Notably, binding motifs for the transcription factor GATA was enriched in ERR occupation regions that exhibited decreased H3K27ac deposition in the context of ERRγ deficiency (Fig. 2b), and decreased ATAC-seq signals in ERRα/γ KO hiPSC-CMs (Fig. 4h). GATA4 has been shown previously to be a key component in cardiac SE in cardiomyocytes[19,26,27]. We next intersected a published GATA4 ChIP-seq dataset (GSE85631) with our ERRγ ChIP-seq to identify putative shared target sites. 18.9% of GATA4 peaks (4460/23,598 peaks) overlapped with ERRγ specific peaks and 13.4% of ERRγ peaks (4460/33,253 peaks) overlapped with GATA4 peaks (Fig. 5a). In sites enriched with both ERRγ and GATA4 peaks (ERRγ + GATA4), the binding motifs for essential cardiac transcription factors such as TEAD and TBX20 were specifically enriched along with the GATA binding site, while these motifs were not found in ERRγ without GATA4 peaks (ERRγ-GATA4, Supplementary Fig. 6a). Notably, 99/213 (46.5%) SE sites contained both ERRγ and GATA4 peaks (SEs + ERRγ + GATA4), and in nearly half of these SE (45/99 sites), ERRγ and GATA4 peaks were colocalized within 200 base pairs (Fig. 5b and Supplementary Fig. 6b). This finding was further supported by analysis of the Cistrome Data Browser[28,29] which allows delineation of similarity between genomic datasets. ERR binding sites denoted as *ESRRA* were confirmed in both ERRγ + GATA4 (GIGGLE score: 973.4) and ERRγ-GATA4 (GIGGLE score: 1093.8) with Cistrome Data Browser as shown in Fig. 5c. This analysis also revealed that ERRγ + GATA4 sites significantly overlapped with the published *MED1*, *EP300*, and *BRD4* cistrome datasets (Fig. 5c), all well-known markers for SEs[30,31]. ERRγ-GATA4 sites often overlapped general enhancer marks including *EP300* and *POLR2A* (Fig. 5c)[32]. GATA4 without ERRγ peaks (GATA4-ERRγ) overlapped MED1 datasets as reported[19] but not other enhancer marks (Supplementary Fig. 6c). In addition, the cistrome datasets of *MEF2A*, *CEBPA*, *SIRT6*, *TAL1*, *STAT2*, *TBLXR1*, *SMARCA4*, *MBD2*, *RUNX1*, and *ORC2* also significantly overlapped with the ERRγ + GATA4 sites (Fig. 5c). In contrast, ERRγ-GATA4 sites did not exhibit this latter trend (Fig. 5c). MEF2 binding sites were often found near the overlapped sites of ERRγ and H3K27ac (Fig. 2b; Supplementary Fig. 3d), and open chromatin sites regulated by ERRα/γ (Fig. 4h). Notably, MEF2 binding has been confirmed on mouse cardiac enhancers[27,33]. CCAAT enhancer-binding protein α (C/EBP-α) coded in *CEBPA* has also been implicated as a transcription factor to drive cardiomyocyte maturation[34], suggesting that MEF2A and C/EBP-α could be potential co-regulators for ERRγ and GATA4 at cardiac enhancers.

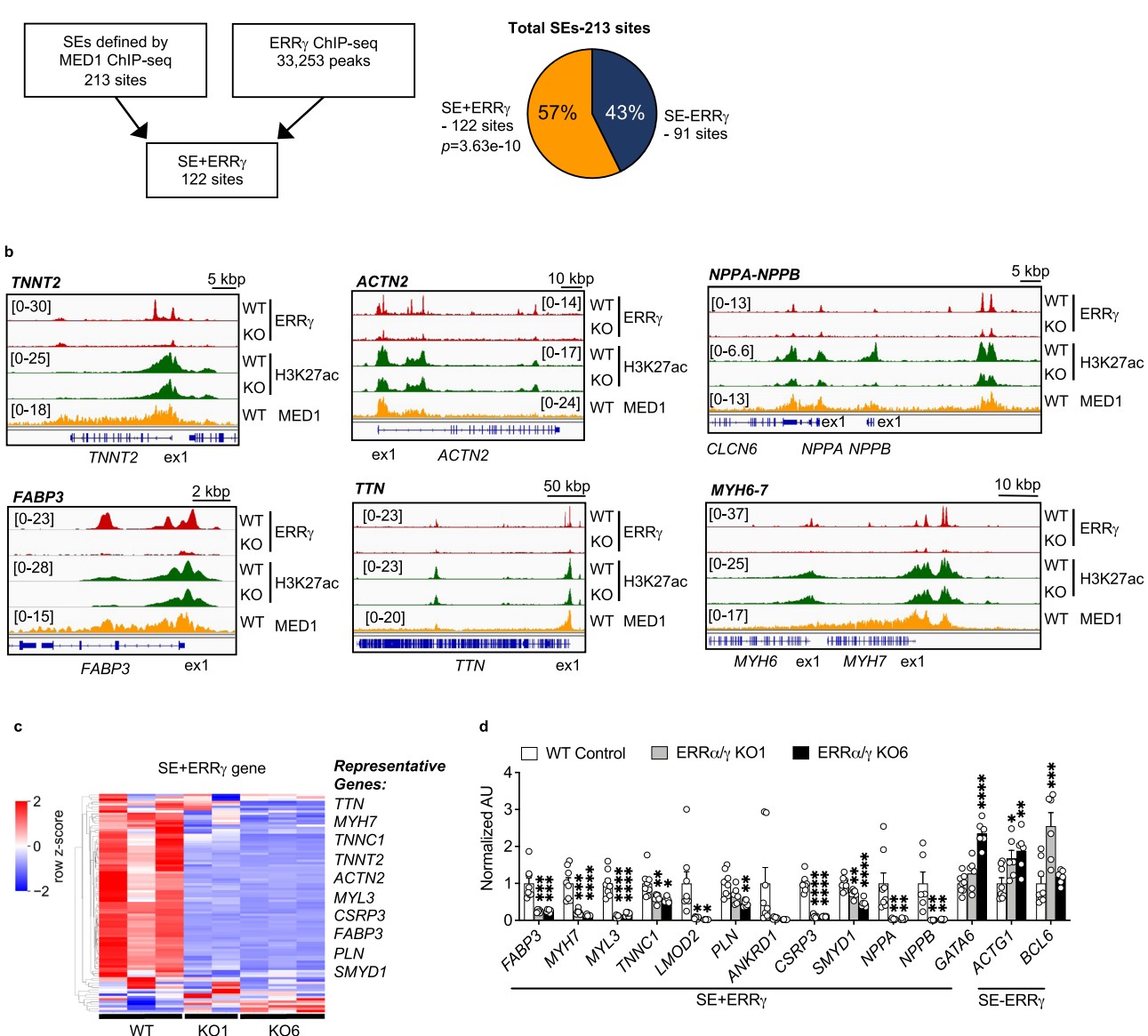

**Fig. 3 ERRγ occupies super-enhancer regions linked to cardiac-enriched structural and metabolic genes in hiPSC-CMs. a** Schematic indicates the intersection analysis with published MED1 chromatin immunoprecipitation sequencing (ChIP-seq; GSE85631) and ERRγ ChIP-seq (GSE113784). Pie chart shows the percent of significantly overlapped sites (Fisher's exact test) from ERRγ ChIP-seq peaks and super-enhancers (SEs) defined by the published MED1 ChIP-seq (GSE85631). **b** Genome browser track of ERRγ (GSE113784), H3K27ac (GSE165965), and MED1 ChIP-seq (GSE85631) signals at indicated loci. **c** Heatmap represents the z-score of expression changes of SE + ERRγ genes in ERRα/γ knockout (KO) human induced pluripotent stem cell-derived cardiomyocytes (hiPSC-CMs). Representative genes that are significantly downregulated in both KO lines are listed next to the heatmap. The row-wise z-scores of fragments per kilobase of exon per million reads mapped (FPKM) values of the presented genes are provided in Supplementary Table 1. **d** Bar graphs represent mRNA expression levels of genes associated with SE + ERRγ or SE determined by real-time quantitative polymerase chain reaction (RT-qPCR) in wild-type (WT)-Control ($n = 8$ or $n = 6$ for *NPPB*) and two different lines of ERRα/γ KO hiPSC-CMs (KO1 and KO6, $n = 6$). *$p < 0.05$, **$p < 0.01$, ***$p < 0.001$, ****$p < 0.0001$, one-way ANOVA followed by Dunnett's multiple comparison test. All bars represent the means ± SEM. $n$ denotes independent biological replicates.

The ERRγ + GATA4 peaks associated with a significant number of genes encoding cardiac structural and ion transport proteins, but were notable for a lack of classical ERR targets involved in metabolism and mitochondrial function (Fig. 5a and Supplementary Data 2). Gene Ontology (GO) terms related to cardiac contractile function-related pathways were highly enriched with the ERRγ + GATA4 sites (Fig. 5d) and none of mitochondrial metabolic pathways were significantly enriched (Fig. 5d). This difference is further exemplified in Fig. 5e, showing that ERRγ and GATA4 peaks colocalized on *TNNI3, MYBPC3,*

and *MYH6-7* enhancer[21] loci but not the *CPT1B* locus (Supplementary Fig. 6d). These results strongly suggest that ERRγ cooperates with GATA4 to activate cardiac-enriched genes involved in contractile processes.

The functions and gene targets of GATA6 are partially redundant with GATA4[35–37]. To determine whether ERRγ and GATA6 also shared targets, we conducted an intersection analysis with GATA6 ChIP-seq in hiPSC-CMs[18] and our ERRγ ChIP-seq datasets. This analysis indicated that only a very small number 2.4% of GATA6 peaks (1355/56,572 peaks; 2.4%) overlapped

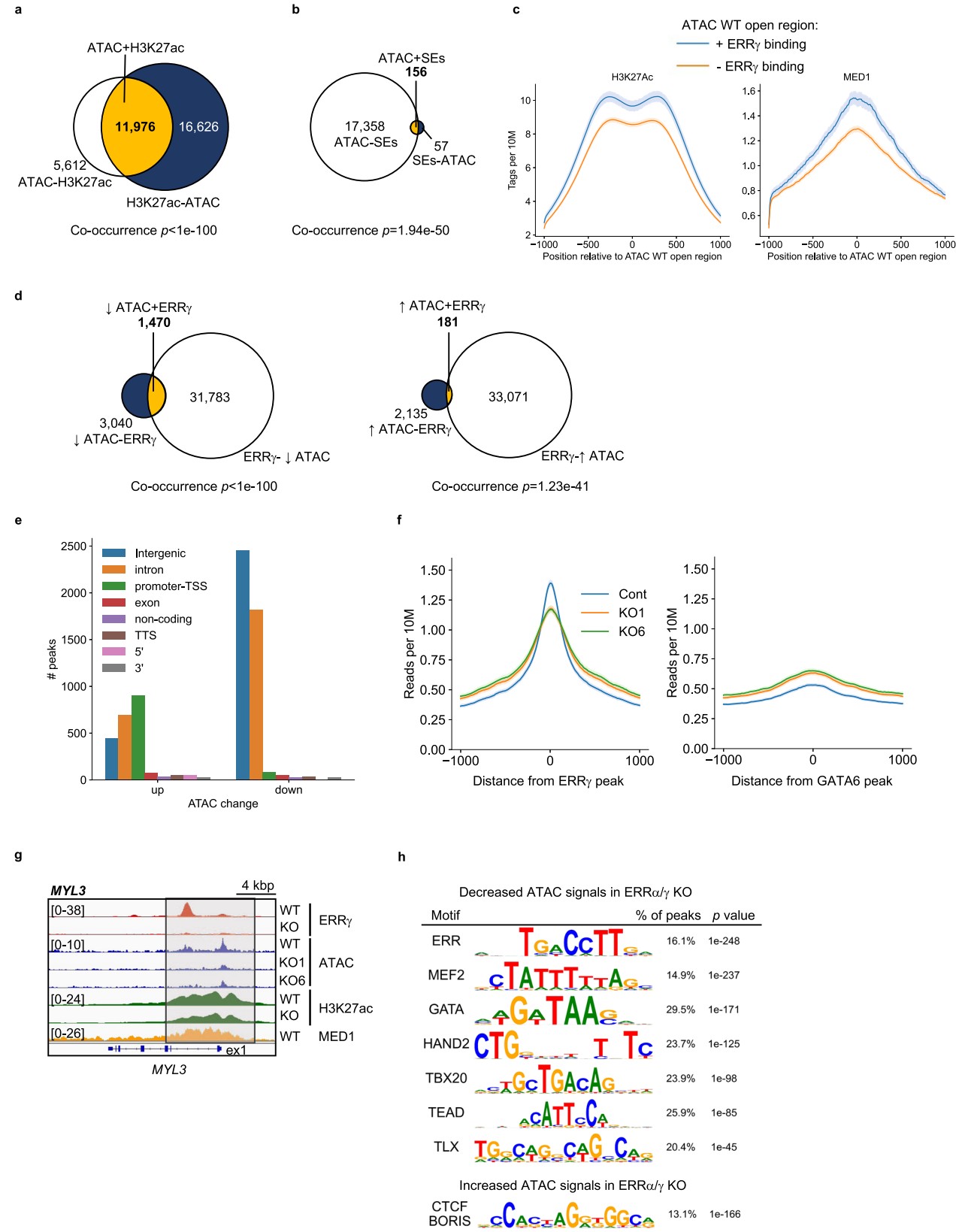

ERRγ peaks (Supplementary Fig. 6e). None of GO terms were significantly enriched with ERRγ and GATA6-shared peaks (ERRγ + GATA6), although a few cardiac genes including *MYL2*, *BIN1*, *KCNQ1*, *ATP2A2*, and *ACTA1* were identified as potential targets (Supplementary Fig. 6e). These results suggest that ERRγ cooperates selectively with GATA4 in the cardiomyocyte.

We next examined the impact of GATA4 loss-of-function on ERR target gene expression. Published GATA4 target genes determined in hiPSC-CMs [19, GSE85631] include cardiac-enriched structural genes involved in the cardiac sarcomere (*TNNI3*, *MYBPC3*, *MYH7*, *TNNT2*, and *TTN* etc.), $Ca^{2+}$ handling (*ATP2A2*, *RYR2*, and *PLN*), and ion transport (*KCNQ1*,

**Fig. 4 ERR regulates cardiac chromatin accessibility in hiPSC-CMs.** Venn diagrams represent the overlapped sites (yellow) between assay for transposase accessible chromatin (ATAC) peaks (GSE165962) and **a** H3K27ac chromatin immunoprecipitation sequencing (ChIP-seq; GSE165965) or **b** published super-enhancer regions (SEs; GSE85631) in wild-type control (WT) human induced pluripotent stem cell-derived cardiomyocytes (hiPSC-CMs). *p*-values were calculated using Fisher's exact test. **c** Aggregation plots for H3K27ac ChIP-seq (GSE165965) and published MED1 ChIP-seq (GSE85631) on high accessible regions with or without ERRγ peaks in WT hiPSC-CMs. The mean tag count and 95% confidence interval are plotted for each position. **d** Venn diagram represents the significantly overlapped sites (Fisher's exact test, yellow) between decreased or increased ATAC peaks (GSE165962) in ERRα/γ knockout (KO) hiPSC-CMs and ERRγ ChIP-seq (GSE113784) in hiPSC-CMs. **e** Bar graphs represent the peak count of ATAC-seq signals located in indicated genomic regions. **f** Aggregation plots for ATAC-seq signals in WT control (Cont) and two ERRα/γ KO hiPSC-CMs (lines 1 and 6) around ERRγ peaks or GATA6 peaks. The mean tag count and 95% confidence interval are plotted for each position. **g** Genomic browser track around *MYL3* containing ERRγ (GSE113784), ATAC (GSE165962), H3K27ac (GSE165965), and MED1 (GSE85631) peaks in hiPSC-CMs. **h** Motif analysis of decreased or increased ATAC-seq signals by KO ERRα/γ in hiPSC-CMs. *p* values were calculated with Fisher's exact test.

*KCNN2*, and *CACNA1C*)[19]. We independently validated these published RNA-seq data in siRNA-mediated GATA4-depleted hiPSC-CMs (Fig. 5f and Supplementary Fig. 6f). Specifically, expression of a subset of the putative ERRγ + GATA4 targets defined by our intersection analysis discussed above were downregulated in GATA4-depleted hiPSC-CMs without changes in the level of *ESRRG* expression (Fig. 5f). In contrast, expression of ERRγ metabolic targets predicted to be independent of GATA4 regulation, such *FABP3* and *CPT1B*, was upregulated in the context of GATA4 deficiency (Fig. 5f). In addition, ERRγ occupation on several ERR + GATA targets including *TNNI3* and *MYH6-7* enhancer[21] was modestly but significantly reduced by GATA4 depletion in hiPSC-CMs (Supplementary Fig. 6g), suggesting that GATA4 occupation is required for ERRγ recruitment on ERRγ + GATA4 targets. In contrast, GATA4 deficiency did not impact ERRγ occupation on the FAO gene *CPT1B*.

To further probe the cooperativity of ERRγ and GATA4 on a shared target gene, we focused on *TNNI3*, an adult cardiac sarcomeric gene isoform and known cardiac maturation marker in human cardiac myocytes[38]. GATA4-mediated induction of *TNNI3* expression was abolished by ERRα/γ deficiency in hiPSC-CMs (Fig. 6a). In addition, the ERR coactivator PGC-1α enhanced the GATA4-mediated regulation of *TNNI3* (Fig. 6a). The PGC-1α-mediated changes in TNNI3 were confirmed at the protein level, while protein levels of TNNI1, a fetal isoform, were not affected by OE of PGC-1α in WT or ERRα/γ KO hiPSC-CMs (Fig. 6b). The recruitment of overexpressed hemagglutinin (HA)-tagged GATA4 (GATA4-HA) on the *TNNI3* locus was significantly decreased by ERRα/γ KO (Fig. 6c). GATA4-HA occupation was also modestly reduced on the *MYH6-7* enhancer[21] in ERRα/γ KO hiPSC-CMs (Fig. 6c). Conversely, this effect was not seen on *CPT1B*, a well-known FAO gene or *POU5F1* used as a negative control (Fig. 6c).

The functional cooperation between ERRγ and GATA4 was further confirmed with cotransfection experiments using a *TNNI3* promoter-luciferase reporter (*TNNI3*-luc) in H9c2 myoblasts. *TNNI3*-luc was synergistically activated by ERRγ and GATA4, an effect that was further enhanced by PGC-1α (Fig. 6d) and blunted by CRISPR/Cas9-mediated ERRα/γ depletion (Fig. 6e). In contrast, the ERRγ and GATA4 cooperation was not observed with *COX6A2*, a metabolic gene promoter, that was identified by our analysis as a GATA4-independent ERR target, although the ERR and PGC-1α cooperation was confirmed as expected (Fig. 6e). Cotransfection studies with *TNNI3* promoter reporters carrying deletion mutation of several ERR response elements (ERRE) or GATA binding sites (Fig. 6f) demonstrated that DNA binding of ERR and GATA factors are essential for full cooperative activation of the *TNNI3* promoter (Fig. 6g). A reporter study with GATA4 zinc finger mutants further confirmed that GATA4 DNA binding is critical for ERRγ/GATA4 cooperation. Deletion of the GATA4 C-terminal zinc

finger region which is essential for DNA binding[39], significantly suppressed the *TNNI3* promoter activation by ERRγ and GATA4, but not an N-terminal zinc finger deletion mutant (Fig. 6h). GATA4 and ERRγ co-binding was also found at the SE region of the the *MYH6-7* cluster (≈7 kb upstream of *MYH7*;[21] Fig. 5e). The *MYH6-7* enhancer-luc reporter was highly activated by the transient OE of ERRγ, GATA4, and PGC-1α (Fig. 6i). These collective results indicate that ERRγ cooperates with the cardiogenic transcription factor GATA4 while bound to DNA to regulate genes involved in cardiac-specific processes such as contractile function, whereas ERRγ-mediated control of energy metabolism occurs independent of GATA4 in hiPSC-CMs. In addition, the results shown here and our previous findings[2,3,40] indicate that PGC-1α serves as a coactivator of both processes.

**PGC-1α/ERRγ form a transcriptional complex with GATA4.** The observed functional interaction between ERRγ and GATA4 suggested that the factors may physically interact. As an initial step to address this possibility, we expressed an ERRγ-Gal4 DNA-binding domain fusion protein (ERRγ-Gal4) and corresponding reporter (pG5luc) in a non-cardiac cell (AD-293) that is void of GATA4. Expression of GATA4 by transfection significantly activated ERRγ-Gal4, but only in the presence of PGC-1α (Fig. 7a). GATA4 failed to activate the ERRγ-Gal4 system in the presence of a PGC-1α LXXLL mutant that abolishes the PGC-1α/ERR interaction[8,9,41] (Fig. 7a). Lastly, targeted point mutations (L449A-F450A or M453A-L454A)[42] of the ERRγ activation function 2 (AF2) domain which is required for interaction with PGC-1α[43], disrupted the ERRγ/GATA4/PGC-1α cooperation (Supplementary Fig. 7a). These results suggested that PGC-1α is essential for the ERRγ and GATA4 cooperation.

We also explored the potential interaction of ERRα with GATA4 using an ERRα-Gal4 construct. These experiments demonstrated a potent cooperative interaction between ERRα, and GATA4 in a PGC-1α-dependent manner similar to that of ERRγ (Supplementary Fig. 7b). Parallel Gal4 DNA-binding domain reporter studies demonstrated that GATA6 also cooperated with ERRα or ERRγ, but that the ERRα and GATA6 cooperativity was significantly weaker than that of ERRα and GATA4 (Supplementary Fig. 7b). These latter results suggest that whereas GATA6 is capable of interacting with ERRγ or ERRα in this experimental context, the ChIP-seq intersection data (Supplementary Fig. 6e) suggests this is not likely a significant interaction, at least in cardiomyocytes.

The importance of PGC-1α for activation of ERRγ + GATA4 target genes including *TNNI3* was further examined by depleting expression of *PPARGC1A* (encoding PGC-1α) using CRISPR interference in hiPSC-CMs. Despite the fact that depletion of *PPARGC1A* expression did not affect mRNA expression levels of *ESRRA*, *ESRRG*, and *GATA4*, the expression levels of genes encoding cardiac structural protein ERR + GATA4 targets including *TNNI3*, *TNNC1*, and *MYBPC3* were significantly

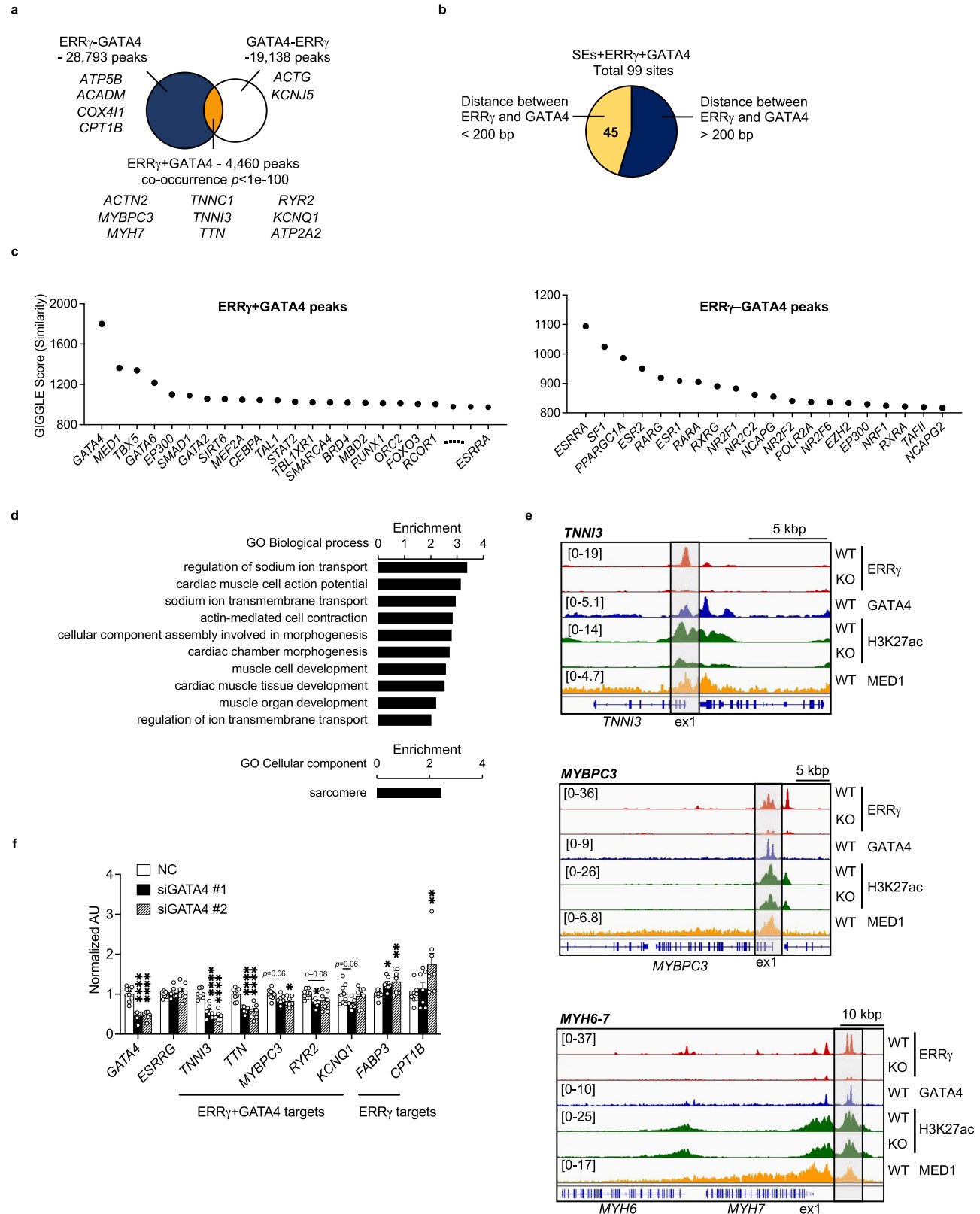

downregulated (Fig. 7b). *PPARGC1A* deficiency also resulted in downregulation of most GATA4-independent ERR targets involved in mitochondrial metabolism such as *ATP5B*[44] and *ACADM*[45] (Fig. 7b). Moreover, depletion of *PPARGC1A* expression resulted in a significant reduction in ERRγ occupation on cardiac-enriched target regulatory regions and metabolic genes

(Fig. 7c). These results indicate that PGC-1α coactivates ERR-mediated activation of both GATA4-dependent and GATA4-independent genes, and that PGC-1α is necessary for ERRγ occupation on sites related to GATA-dependent target regulation.

To further assess the physical interaction of ERRγ and GATA4, immunoprecipitation (IP) studies were conducted in hiPSC-CMs.

**Fig. 5 ERRγ cooperates with GATA4 to activate the transcription of genes coding cardiac structural proteins in hiPSC-CMs. a** Venn diagram indicates the overlapped peaks from published GATA4 chromatin immunoprecipitation sequencing (ChIP-seq; GSE85631) and ERRγ ChIP-seq (GSE113784) datasets generated from human induced pluripotent stem cell-derived cardiomyocytes (hiPSC-CMs). GATA4 and ERRγ-shared peaks (ERRγ + GATA4) are presented by orange-colored region. $p$ values were calculated using Fisher's exact test. **b** Pie chart represents the overlapped sites from published GATA4 ChIP-seq, cardiac super-enhancers (SEs; GSE85631), and ERRγ ChIP-seq (GSE113784) datasets in hiPSC-CMs. The 45 SEs where ERRγ and GATA4 peaks are colocalized within 200 base pairs are highlighted with yellow. **c** Top 20 transcription factors that significantly overlapped with ERRγ + GATA4 and ERRγ-GATA4 are plotted. *ESRRA* was ranked in the top 30 in the ERRγ + GATA4 analysis. Y axis represents the similarity (GIGGLE score) between published datasets and each peak set. **d** Bars indicate enrichment score for Gene Ontology (GO) Biological Process and GO Cellular Component terms significantly enriched in ERRγ + GATA4 targets. **e** Representative genomic browser tracks of *TNNI3*, *MYBPC3*, and *MYH6-7* regions. **f** Bar graphs represent mRNA expression levels of a subset of ERRγ + GATA4 and ERRγ targets in Negative Control (NC) ($n = 9$) and siGATA4#1 ($n = 7$) or siGATA4#2 ($n = 7$) transfected hiPSC-CMs. $*p < 0.05$, $**p < 0.01$, $****p < 0.0001$, one-way ANOVA followed by Dunnett's multiple comparisons test. All bars represent the means ± SEM. $n$ denotes independent biological replicates.

An interaction between overexpressed FLAG-tagged ERRγ and endogenous GATA4 was observed in hiPSC-CMs (Fig. 7d). To explore the interaction of ERRγ/GATA4 with the coactivator PGC-1α, IP was performed with epitope-tagged ERRγ (3xFLAG), PGC-1α (Myc/His), and GATA4 (HA) in AD-293 cells (Supplementary Fig. 7c). IP with antibodies to FLAG or HA epitope tags again confirmed the ERRγ/GATA4 interaction as well as the expected interaction of PGC-1α and ERRγ (Supplementary Fig. 7c). However, a GATA4 and PGC-1α interaction was not observed, suggesting that ERRγ may serve as a hub for these two factors (Supplementary Fig. 7c). These results, together with that of the ChIP studies and Gal4 DNA-binding domain assays, support a mechanism in which GATA4 and ERRγ physically interact while co-occupying DNA enhancer regions that regulate cardiac-enriched structural genes, and that PGC-1α serves to enhance the function of this complex via its canonical interaction with ERRγ.

**Assessment of the ERR circuitry in human heart disease**. Several naturally occurring GATA4 mutations have been reported to cause human congenital heart disease[46,47]. The ERRγ-Gal4 system was used to determine the functional impact of naturally occurring GATA4 mutants on ERRγ and GATA4 cooperativity. Two mutations (G296S and G296C) attenuated the ERRγ, PGC-1α, and GATA4 cooperation (Fig. 7e). In addition, the G296S mutation compromised ERRγ/GATA4-mediated activation of the *TNNI3* promoter (Fig. 7f). Importantly, this mutation has been known to decrease GATA4 DNA binding[47], further supporting our conclusion that GATA4 DNA binding is essential for full ERRγ and GATA4 cooperativity (Fig. 6g, h). The interaction of ERRγ/GATA4/PGC-1α was not significanly disrupted with GATA4 G296S (Supplementary Fig. 7d), suggesting that the mutation alters function but not the physical interaction of this complex. Notably, the G296S mutant has been linked to atrial and ventricular septal defects as well as cardiomyopathy[19]. In addition, mice engineered for the G295S mutation (which corresponds to the human G296S mutation) develop a ventricular non-compaction phenotype[48]. Interestingly, we have observed a similar left ventricular non-compaction phenotype in ERRα/γ-deficient mice[10]. Comparison of the RNA-seq datasets (Fig. 7g) generated from ERRα/γKO and GATA4 G296S (GSE85631) hiPSC-CMs (vs. corresponding WT hiPSC-CMs) demonstrated a shared subset of dysregulated sarcomere genes such as *TNNI3*, *CSRP3*, *LMOD2*, *TCAP* and *MYH7* (Fig. 7g). These results suggest that alteration in ERRγ and GATA4 cooperativity may contribute to the G296S mutant phenotype.

We next sought to determine if ERR signaling is dysregulated in acquired forms of human heart failure. To this end, we conducted a comparative intersection of our RNA-seq data from ERRα/γ KO hiPSC-CMs with a dataset generated from right ventricular samples of humans with heart failure with reduced

ejection fraction (HFrEF) compared to normal ventricular function controls[49]. Levels of *ESRRA*, *ESRRG*, and *PPARGC1A* were significantly downregulated (Supplementary Fig. 8a), although *GATA4* level was not significantly regulated (Supplementary Fig. 8a). The gene expression data suggest that ERR/PGC-1α axis is dysregulated in HFrEF. 24.9% of downregulated genes in ERRα/γ KO hiPSC-CMs were also significantly downregulated in HErEF (376/1509 genes, Supplementary Fig. 8b). The corresponding enrichment analysis suggests that the dysregulated ERR/PGC-1α axis results in deactivated mitochondrial metabolism in HFrEF (Supplementary Fig. 8c). In addition, levels of genes encoding cardiac structural components (*TNNT2*, *TCAP*, *TMOD1*, *TTN*, and *MYL3*), ion transporters (*KCNJ8*, *KCNK3*, and *ATP1B1*), and $Ca^{2+}$ handling protein (*ATP2A2*) were commonly downregulated in ERRα/γ KO hiPSC-CMs and HFrEF, resulting in the enriched Kyoto Encyclopedia of Genes and Genomes (KEGG) pathway term related to cardiac muscle contraction (Supplementary Fig. 8c). Among them, *TTN*, *KCNJ8*, *KCNK3*, *ATP1B1*, and *ATP2A2* were defined as ERRγ + GATA4 targets (Supplementary Data 2). Taken together, these findings and our hiPSC-CM studies indicate that ERR/PGC-1α axis is an essential transcriptional circuitry to maintain both mitochondrial oxidative metabolism and cardiac contractile function in human adult hearts, and that GATA4 cooperation is involved in the latter program. Dysregulation of the ERR/PGC-1α/GATA4 transcription factor complex in the failing heart could contribute to the well-described fetal shifts characterized by reduced expression of genes involved in adult mitochondrial oxidative metabolism, contractile function and ion transport including calcium handling.

## Discussion
Recently, we found that the nuclear receptors, ERRα and γ, are essential for normal postnatal cardiac developmental maturation[10]. This observation set the stage for the studies described herein aimed at understanding how the nuclear receptor ERR coordinately regulates canonical gene targets involved in mitochondrial energy metabolism, a function that is relevant to most cells and tissues, with cardiac-specific processes such as sarcomeric function and ion transport. Our genomic interrogation studies demonstrated that ERRγ is a functional component of many promoters, cardiomyocyte enhancers, and SE, a subset of which involved colocalization with the cardiogenic transcription factor GATA4. We found that ERRγ regulates cardiac-specific gene targets in cooperation with GATA4, whereas its transcriptional control of genes involved in mitochondrial processes occurs largely independent of GATA4. Importantly, both processes are coregulated by PGC-1α suggesting a mechanism for upstream coordination of these transcriptional regulatory circuits. Lastly, the functional cooperation of ERRγ/GATA4/PGC-1α is diminished by the human disease-causing

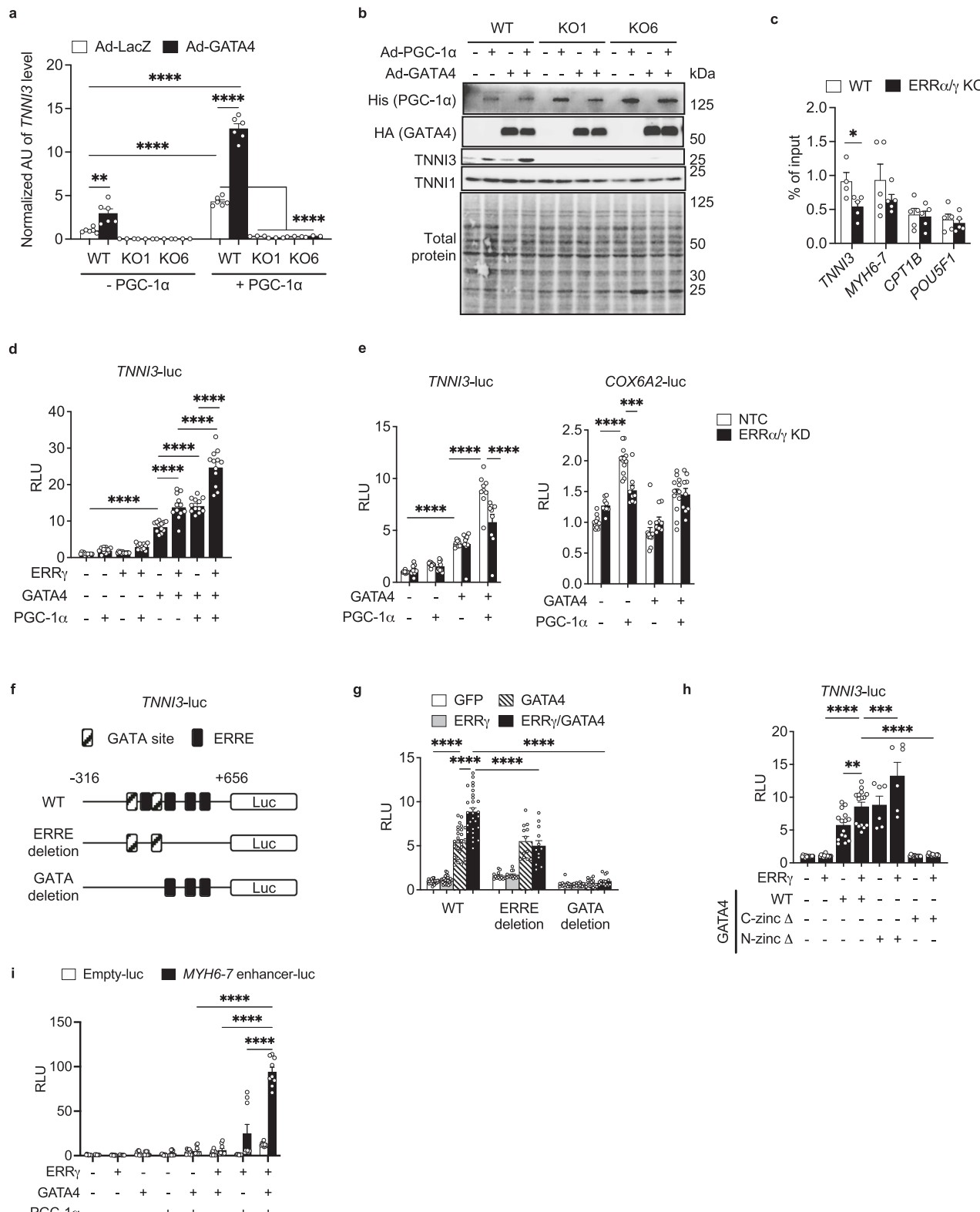

mutation GATA4 G296S and expression of downstream target genes was reduced in failing human heart samples. Together these results identify a mechanism whereby mitochondrial ATP-producing capacity is coordinately regulated with cell-specific energy-consuming processes via cooperativity between a nuclear receptor, a cardiogenic transcription factor, and a common cor-egulator (Fig. 8).

In this study, we used the hiPSC-CM system to probe the mechanisms whereby ERR signaling regulates diverse cardiomyocyte maturation processes. We found that ERRα/γ loss-of-function reduces expression of metabolic and structural cardiomyocyte maturation markers. It is well-known that hiPSC-CMs exhibit an immature phenotype that resembles fetal cardiomyocytes[50]. Therefore we repeated our studies using a recently described

**Fig. 6 ERR and GATA4 activate *TNNI3* transcription in the presence of PGC-1α. a** mRNA expression level of *TNNI3* in wild-type (WT) control (*n* = 6) and ERRα/γ knockout (KO) line 1 or 6 (*n* = 3) human induced pluripotent stem cell-derived cardiomyocytes (hiPSC-CMs) following the overexpression of PGC-1α and/or GATA4 (Ad-PGC-1α and/or Ad-GATA4). **\*\****p* < 0.01, **\*\*\*\****p* < 0.0001, two-way ANOVA followed by Tukey's multiple comparisons test. **b** Representative immunoblot images to show hemagglutinin (HA)-tagged GATA4, His-tagged PGC-1α, TNNI3, and TNNI1. **c** Levels of HA-tagged GATA4 occupation on the indicated targets in WT (*n* = 5 or *n* = 4 for *TNNI3* and *POU5F1*) and ERRα/γ KO hiPSC-CMs (*n* = 5). \**p* < 0.05 vs WT, two-tailed student's *t*-test. *MYH6-7* denotes the enhancer site of *MYH6-7* cluster. **d** Bar graphs represent relative light units (RLU) derived from *TNNI3* promoter-luciferase reporter (*TNNI3*-luc) with overexpression of ERRγ, GATA4, and PGC-1α in H9c2 myoblast. of *TNNI3*-luc. *n* = 12, **\*\*\*\****p* < 0.0001, one-way ANOVA followed by Tukey multiple comparisons test. **e** Bar graphs represent the RLU derived from *TNNI3*-luc or *COX6A2* luciferase reporter (*COX6A2*-luc) in H9c2 myoblasts transduced with CRISPR/Cas9 and non-targeting control (NTC) guide RNA (gRNA) or gRNA targeting ERRα and γ (ERRα/γ KD). *n* = 12 for NTC in *COX6A2*-luc or *n* = 9 for others. **\*\*\****p* < 0.001, **\*\*\*\****p* < 0.0001, two-way ANOVA followed by Tukey's multiple comparisons test. **f** Schematic of *TNNI3* promoter with putative ERR response elements (ERRE) and GATA binding sites. **g** WT (*n* = 25), ERRE-deleted (*n* = 12) or GATA binding sites-deleted (*n* = 16) -*TNNI3*-luc were assessed with overexpression of ERRγ and/or GATA4 in H9c2 myoblasts. \**p* < 0.05, **\*\*\*\****p* < 0.0001, two-way ANOVA followed by Tukey's multiple comparisons test. **h** Bar graphs represents RLU derived from *TNNI3*-luc with overexpression of ERRγ (*n* = 15), GATA4 WT (*n* = 15), C-teriminal zinc finger deleted GATA4 (C-zincΔ, *n* = 12), N-terminal zinc finger deleted GATA4 (N-zincΔ, *n* = 6), the combination of ERRγ and WT (n = 15) or each GATA4 mutant (C-zincΔ, *n* = 12; N-zincΔ, *n* = 6) in H9c2 myoblasts. \**p* < 0.05, **\*\*\*\****p* < 0.0001, one-way ANOVA followed by Tukey's multiple comparisons test. **i** Bar graphs represents RLU derived from control reporter (empty-luc; *n* = 8 for ERRγ + PGC-1α or *n* = 9 for others) and *MYH6-7* enhancer-luc (*n* = 9) with overexpressed indicated factors in AC16 cells. **\*\*\*\****p* < 0.0001, two-way ANOVA followed by Tukey's multiple comparisons test. All bars represent the means ± SEM. *n* denotes independent biological replicates.

maturation cocktail[15]. Using this system, we again found that ERRα/γ are necessary for induction of adult cardiomyocyte metabolic and structural maturation genes (Fig. 1f–g). Notably, recent studies by others have implicated ERR in hiPSC-CM maturation. For example, an hiPSC-CM engineered heart model exibited increased expression of *ESRRA* and *PPARGC1A*[51]. In addition, an ERRγ agonist was recently identified as an inducer to drive hiPSC-CM maturation in an unbiased screen using a *TNNI3* reporter hiPSC line[52]. These collective results suggest that strategies aimed at activation of ERR signaling may enhance cardiomyocyte maturation in culture and in vivo.

Our results identify a role for ERRγ in a significant subset of cardiomyocyte enhancers. ERRγ loss of function reduced H3K27ac deposition at a subset of the enhancer regions, and this effect was most pronounced at enhancers associated with activated targets as compared to suppressed targets. We also found that ERRγ functions in a significant number of cardiomyocyte promoters and SEs. Notably, SEs-containing ERRγ were largely associated with genes encoding cardiac-enriched structural proteins including adult contractile isoforms, and often contained GATA4 binding sites. Consistent with our findings, embryonic heart-specific enhancer segments defined recently in human cardiac tissues demonstrated enrichment of binding motifs for GATA, MEF2, and ERRs[53]. In addition, ERRγ occupies the recently described evolutionarily conserved cardiac SE region associated with the cardiac-specific *NPPA-NPPB* gene complex[20] as well as the *MYH6-7* cluster[21]. We also found that ERRγ affects chromatin accessibility at a subset of its enhancer occupation sites. The mechanism for this latter observation is unclear but could be related to direct interactions with chromatin remodeling enzymes. Alternatively, or in addition, indirect mechanisms for ERR-mediated chromatin modulation are possible. To this possibility, we found that the cardiac-enriched histone-lysine methyltransferase, SMYD1, is an ERRγ target. Interestingly, SMYD1 activates PGC-1α expression[54] suggesting the existence of a feedforward activation loop between SMYD1 and the ERR/PGC-1α axis.

A major finding of this work is that ERRγ regulates its cardiac-enriched targets including many sarcomeric genes, in physical and functional cooperation with the cardiogenic transcription factor GATA4. In contrast, we found that ERRγ-mediated transcriptional control of genes involved in mitochondrial processes occurs largely in a GATA4-independent manner, and does not involve SEs. It is possible, however, that this latter mechanism involves additional cardiogenic transcription factors such as MEF2 and TEAD[27], given

that binding motifs for each are often enriched near ERR sites. These results suggest a broader mechanism for the coordinate control of energy metabolism, a ubiquitous process, with cell-specific functions. Consistent with this notion, ERRs have been shown to regulate cell-specific processes in other tissues. In the brown adipocyte, ERRα occupies SE regions associated with the *Ucp1* gene[55] and ERRβ has been shown to localize to enhancers and regulate chromatin accessibility in stem cells[30,56,57]. ERRγ functions with hepatic nuclear factor 1 β (HNF1β) in renal epithelial cells for regulating kidney-specific genes[58] and ERRα has recently been shown to coordinate energy metabolism and differentiation of renal proximal tubule cells[59]. We speculate that ERRs, and perhaps other nuclear receptors, cooperate with tissue-specific transcriptional regulators to coordinately regulate mitochondrial energy production with cell-specific demands such as contractile function or thermogenesis.

Our results suggest an important role for the known ERR coactivator PGC-1α in the integration of the transcriptional control of both GATA-dependent and GATA-independent ERRγ gene targets. PGC-1α is a well-characterized inducible coactivator of the ERRs[8,9]. The IP results indicate that PGC-1α interacts directly with ERRγ as expected, but not GATA4. However, PGC-1α was shown to be necessary for full transcriptional activation for both circuits and enhances ERRγ occupation at the shared GATA4/ERRγ enhancers. We propose that PGC-1 serves as a common integrator of GATA4-dependent and -independent ERR-regulated genes in order to match capacity for cardiomyocyte ATP production and demand. According to this proposed mechanistic model, it is possible that upstream signaling pathways that sense cellular energy status or physiological inputs that determine changes in cardiac work may converge on both ERR/PGC-1α and ERR/GATA4/PGC-1α. Consistent with this notion, PGC-1α has been shown to be regulated by the cellular energy-sensing kinase AMPK[60,61]. The observation that normal DNA occupation and function of ERR on both GATA4-dependent and independent targets complex requires PGC-1α suggests that this coregulator recruits additional stabilizing components of the transcriptional activation machinery and/or chromatin remodeling enzymes. The observed effects of ERRγ deficiency on enhancer H3K27ac occupation is consistent with this notion. In addition, PGC-1α has been shown to interact with the components of the mediator complex[62].

A number of human disease-causing GATA4 mutations have been described[46,47]. The GATA4 G296S mutation, which we found altered ERRγ, GATA4, and PGC-1α cooperativity, has been shown to cause cardiac septal defects and cardiomyopathy[19,47]. In addition,

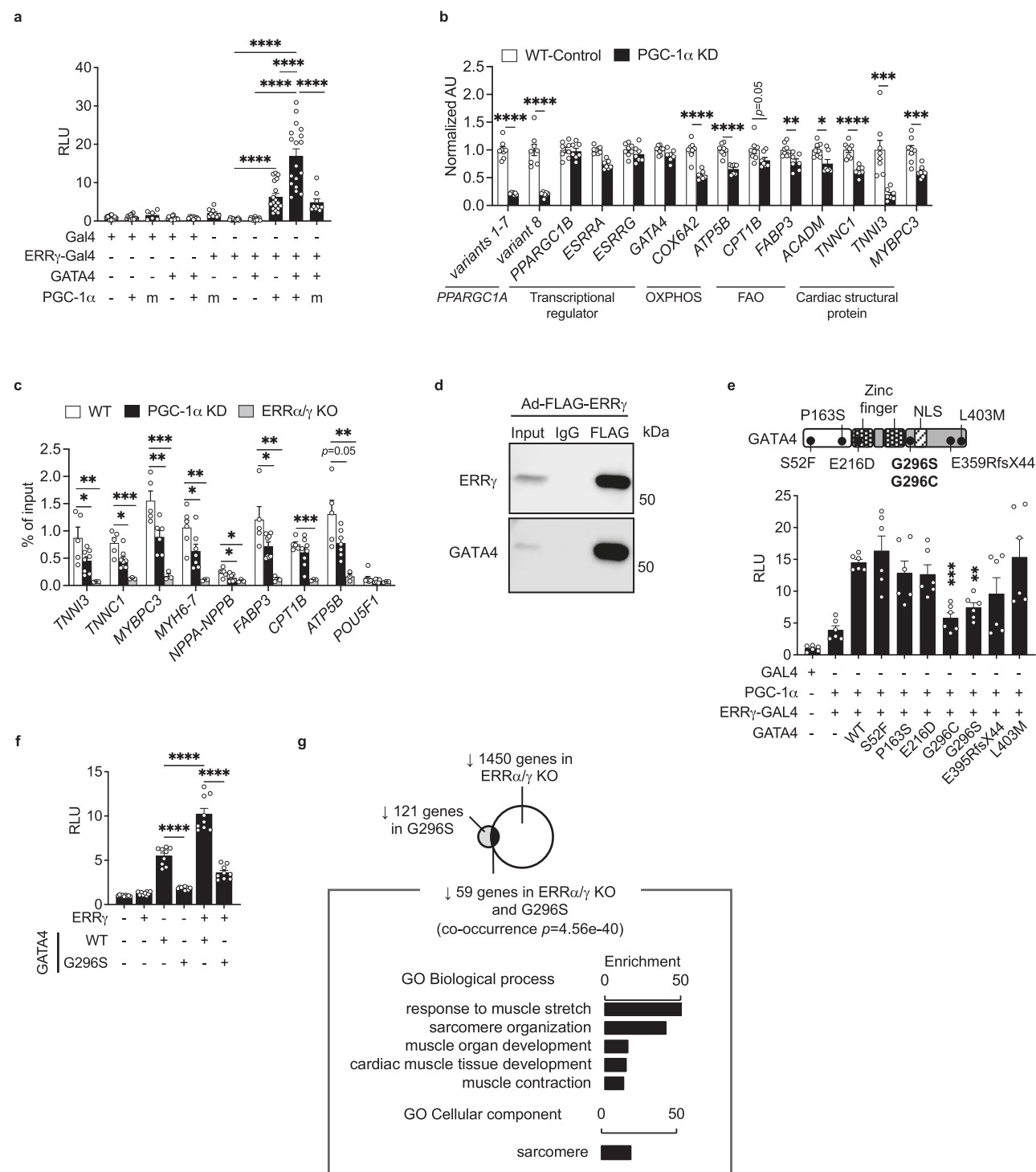

a GATA4 G295S (corresponding to the human G296S mutation) knock-in mouse develops a ventricular non-compaction phenotype. Notably, we have shown cardiac-specific targeted disruption of the *Esrra/Esrrg* genes during fetal development in mice also results in a ventricular compaction phenotype[10]. The GATA4 G296S mutation has been shown to alter cooperation with transcription factor partners such as TBX5 as demonstrated in hiPSC-CM derived from a patient carrying the G296S mutation, resulting in a defective cardiac gene program[19]. We found that hiPSC-CM from a patient with the GATA4 G296S mutation[19] resulted in a significant reduction in the expression of many GATA4 target genes including shared ERR + GATA4 targets. In addition, our results demonstrate

that G296S alters ERRγ and GATA4 cooperativity, although this mutation does not significantly attenuate the ERRγ and GATA4 physical interaction. Our findings suggest that in addition to impacting the interaction of GATA4 with TBX5, the G296S mutation can also alter PGC-1α/ERRγ/GATA4 cooperativity which could contribute to the human disease phenotypes including cardiomyopathy which occurs in ERR-deficient mice[10,11].

To further explore the translational relevance of the PGC-1α/ERR/GATA4 interaction, we compared the gene expression changes in ERRα/γ KO hiPSC-CMs with that of corresponding transcriptomic datasets generated from human HFrEF samples[47]. The expression of ERR, PGC-1α, and a subset of gene targets was

**Fig. 7 ERRγ/PGC-1α interact with GATA4 to form a transcriptional regulatory complex in cardiomyocytes. a** Bar graphs represent the relative light unit (RLU) from pG5luc construct. ERRγ-Gal4, GATA4/PGC-1α/ERRγ-Gal4, GATA4/ERRγ-Gal4, and PGC-1α/ERRγ-Gal4, $n = 18$; Gal4, $n = 17$; PGC-1α/Gal4, $n = 16$; GATA4/Gal4, $n = 15$; GATA4/PGC-1α/Gal4, PGC-1α m/ERRγ-Gal4, PGC-1α m/ERRγ-Gal4, $n = 9$; PGC-1α m/Gal4, $n = 6$, ****$p < 0.0001$, one-way ANOVA followed by Tukey's multiple comparisons test. **b** Bar graphs represent the mRNA levels of indicated genes in wild-type control (WT, $n = 8$) and PGC-1α knockdown (KD, $n = 8$) human induced pluripotent stem cell-derived cardiomyocytes (hiPSC-CMs). *$p < 0.05$, **$p < 0.01$, ***$p < 0.001$, ****$p < 0.0001$ vs WT, two-tailed Student's t-test. **c** Levels of ERRγ occupation on the indicated targets in WT ($n = 6$ for MYH6-7 and NPPA-NPPB, $n = 8$ for POU5F1, $n = 5$ for others), PGC-1α KD ($n = 8$ for TNNI3, TNNC1, FABP3, and CPT1B, $n = 6$ for MYBPC3 and POU5F1, $n = 7$ for MYH6-7, NPPA-NPPB, and ATP5B), and ERRα/γ knockout (KO) hiPSC-CMs ($n = 3$). *$p < 0.05$, **$p < 0.01$, ***$p < 0.001$ vs WT, one-way ANOVA followed by Dunnett's multiple comparisons test. Stem cell enhancer region on POU5F1 was used as a negative control. MYH6-7 and NPPA-NPPB denote the distal enhancer sites upstream of each gene cluster. **d** Representative immunoblot images to show the interaction between FLAG-tagged ERRγ and endogenous GATA4 in FLAG-ERRγ overexpressed hiPSC-CMs. **e** Schematic to indicate reported naturally occurring GATA4 mutations. NLS denotes nuclear localization signal. ERRγ-Gal4 experiment was performed with PGC-1α and GATA4 natural mutants. Bar graphs indicate RLU from pG5luc construct in AD-293 cells. **$p < 0.01$, ***$p < 0.001$, vs ERRγ, PGC-1α, and WT GATA4 transfected group, one-way ANOVA followed by Tukey multiple comparisons test. $n = 6$. **f** TNNI3-luc reporter experiment with overexpression of ERRγ and GATA4 WT or GATA4 G296S in H9c2 myoblast. Bar graphs represent RLU of TNNI3-luc. $n = 9$, ****$p < 0.0001$, one-way ANOVA followed by Tukey multiple comparisons test. **g** Venn diagram showing significant overlap (using Fisher's exact test) between RNA-seq datasets generated in G296S GATA4 hiPSC-CMs (GSE85631) and ERRα/γ KO hiPSC-CMs (GSE165963). Bar graphs represent enrichment score for significantly enriched Gene Ontology (GO) Biological Process and GO Cellular Component with the commonly downregulated genes. All bars in **a, b, c, e**, and **f** represent the means ± SEM. $n$ denotes independent biological replicates.

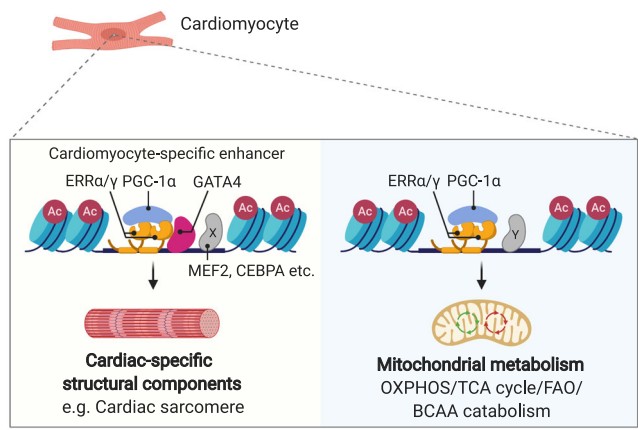

**Fig. 8 Proposed model for coordinate transcriptional control of cardiac energy production and cardiac-specific processes such as the sarcomere by ERRγ.** The figure was created with BioRender.com.

downregulated in the HF samples. Interestingly, both GATA4-independent (energy metabolic) and -dependent (contractile isoforms/ion channels) ERR targets were downregulated in the HF samples consistent with deactivation of the ERR/GATA4 maturation circuit. These results are consistent with the shift toward fetal gene programs involved in both energy metabolism and the contractile machinery known to occur in the failing heart. Taken together with the results of the naturally occurring mutant G296S, we suggest that disruption of the PGC-1α/ERR/GATA4 cooperative interaction could contribute to inherited and acquired human heart disease. Future studies aimed at replicating these results in additional independent human heart disease datasets will be important.

## Methods

**hiPSC culture system.** All experimental protocols have been approved by the office of Environmental Health and Radiation Safety at the University of Pennsylvania. Human male iPSCs [α-Skin; provided by the laboratory of Huei-Sheng Vincent Chen (Indiana University School of Medicine), or WTC11; provided by the laboratory of Deepak Srivastava (Gladstone Institute)], were cultured and maintained in TeSR-E8 (Stem Cell Technologies, #05990). hiPSCs were used between 20 and 60 passages. Generation of both ERRγ and ERRα/γ KO α-Skin lines with CRISPR/Cas9 were previously published[10]. Briefly, pSpCas9n(BB)-2A-Puro [Addgene plasmid # 48141, provided by Dr. Feng Zhang (Massachusetts Institute of Technology)][63] expressing gRNA targeting ESRRG, and pCas9-EGFP and pGuide [both kindly provided by Dr. Kiran Musunuru (University of

Pennsylvania)] expressing gRNA targeting ESRRA were transfected to α-Skin hiPSC with Lipofectamine 3000 (ThermoFisher, L3000008) or Lipofectamine Stem Transfection Reagent (ThermoFisher, STEM00015). The single clones were isolated, and indels at the target loci were confirmed by Sanger sequencing. Deficient protein expression of ERRα and ERRγ in the KO clones was confirmed by western blotting[10]. For lentiviral infection, hiPSC was centrifuged with the media containing lentiviruses at $800 \times g$ for 90 min at room temperature. The media was removed and fresh TeSR-E8 were supplied. Two days after the viral infection, the infected cells were selected with puromycin (0.5 μg/mL) until non-infected cells were killed completely.

**hiPSC-derived cardiomyocyte differentiation.** CDM3 hiPSC-CM differentiation was performed with minor modifications as detailed below[64,65]. All hiPSC lines were differentiated with CDM3 media consisting of RPMI 1640 (11875119, Gibco), 500 μg/mL recombinant human albumin (Sigma-Aldrich, A0237 or Sciencell, OsrHSA), and 213 μg/mL L-ascorbic acid 2-phosphate sesquimagnesium (Sigma-Aldrich, A8960-5G) until day 10. ERRγ KO and its control hiPSC-CMs were maintained with glucose-deficient CDM3 with sodium DL-lactate (Sigma-Aldrich, L4263-100ML) supplementation for 5 days. WTC11 hiPSC-CMs were enriched with glucose-deficient media for 2 days. ERRα/γ KO and control hiPSC-CMs were maintained until day 22 with CDM3 media. We confirmed the cardiac differentiation efficiency in both lines was similar, which was more than 90%[10]. The purified cardiomyocytes were replated on matrigel-coated plates in 10% FBS supplemented with RPMI 1640 media (Thermo Fisher Scientific, 11875119) and 5 μM Y27632 (Selleck Chemical LLC, S104910MG). Two days after replating, the ERRγ KO and its control cardiomyocytes were maintained with the glucose-deficient media for an additional 5 days. siRNA transfection was conducted in WTC11 hiPSC-CMs 2 days after replating as described in siRNA transfection section. Adenoviral infection was performed after the 2nd purification with CDM3 media supplemented with 5% FBS.

**hiPSC-derived cardiomyocyte culture with maturation cocktail.** Day 10 WT or ERRα/γ KO hiPSC-CMs were cultured with the previously reported maturation cocktail[15] for 7 days with minor modification as described below. CDM3 media was supplemented with 4 nM 3,3′,5-triiodo-L-thyronine (Sigma-Aldrich, T5516-1MG), 100 ng/mL dexamethasone (Sigma-Aldrich, D4902-25MG), 1 μM GW7647 (Cayman, 10008613), 200 μM palmitate (P9767, Sigma-Aldrich)-conjugated with fatty acid-free bovine serum albumin (Sigma-Aldrich, A6003-100G), and 500 μM carnitine (C0283-5G, Sigma-Aldrich).

**DNA plasmid construction.** Human ERRγ (ESRRG variant1, NM_001438.3) or ERRα (ESRRA variant1, NM_004451.5) with 3xFLAG tag at N-terminal (3xFLAG-ERRγ or 3xFLAG-ERRα), human GATA4 variant 2 (NM_002052.5) or GATA6 (NM_005257.6) with HA tag at C-terminal (GATA4-HA or GATA6-HA) was amplified by PCR from human cDNA and cloned into pcDNA3.1 (-) (Thermo Fisher Scientific). EGFP from EGFP-N1 (Clontech) was cloned into the XhoI/NotI site in pcDNA3.1 (-). PGC-1 α was cloned into HindIII-digested pcDNA3.1myc/his (Invitrogen) to generate pcDNA3.1 (-)-WT PGC-1α tagged with Myc/His (PGC-1α-Myc/His) and its LXXLL mutant version was generated with the Quickchange Mutagenesis kit (Stratagene)[8,66]. For luciferase promoter assays, human TNNI3, COX6A2, or SMYD1 promoter region was amplified by PCR and cloned from human genomic DNA into pGL3-Basic (Promega). Distal enhancer region at the upstream of MYH6-7 cluster[21] was cloned from human genomic DNA into pGL4.24 (Promega). ERRγ or ERRα was cloned into pCMX-Gal4 plasmid to

generate pCMX-ERRγ or pCMX-ERRα fused with Gal4 DNA-binding domain (ERRγ-Gal4 or ERRα-Gal4)[8]. All PCR and ligation reactions for cloning were conducted with In-Fusion HD cloning plus (Takara, 638911). In-Fusion HD cloning kit was also employed to generate TNNI3 promoter mutant reporters (ERRE-deletion and GATA binding sites-deletion mutants), ERRγ AF2 mutants-Gal4 (L449A-F450A and M453A-L454A) or GATA4 mutants (N-zincΔ, C-zincΔ, S52F, P163S, E216D, G296C, G296S, E395RfsX44, and L403M). JASPAR (http://jaspar.genereg.net/)[24,25] was used to determine putative ERRE and GATA binding motifs on TNNI3 promoter region with default setting. Defined sites were illustrated in Fig. 6f. All of the cloning and mutagenesis primer sets are listed in the Supplementary Tables 2 and 3.

**siRNA transfection**. One hundred nanomolar Dicer-Substrate Short Interfering RNAs (Integrated DNA Technologies, hs.Ri.GATA4. 13.1 or 13.2) targeting human GATA4 or negative control (Integrated DNA Technologies, 51-01-14-03) were transiently transfected with lipofectamine RNAi max (ThermoFisher) according to manufacturer's protocol in WTC11 hiPSC-CMs around Day 15. Three days after the siRNA transfection, the cells were harvested and processed for gene expression or chromatin IP studies.

**Generation of adenovirus for ERRγ or GATA4 overexpression**. Human ESRRG variant1 (NM_001438.3) with FLAG tag at N-terminal was cloned by Platinum™ Taq DNA Polymerase High Fidelity (Thermo Fisher Scientific) into Bgl II and blunted XhoI sites of pAdTrack-CMV. Adenoviral vectors expressing FLAG-tagged ESRRG, Ppargc1a[2], and control vector (empty vector) were generated by the AdEasy system[67]. Human GATA4 variant 2 (NM_002052.5) with HA tag at C-terminal was cloned by CloneAmp HiFi PCR Premix (Takara, 639298) into pAdenox-CMV (Takara, 632269). For control adenovirus, LacZ was cloned into the same backbone. The primer sets for cloning are listed in Supplementary Table 2. Lineared adenoviral vectors were transfected into AD-293 (Agilent, 240085) with lipofectamine 2000 (ThermoFisher, 11668019) to amplify the adenoviruses. The virus titer was measured by Adeno-X RapidTiter Kit (Takara, 632250).

**Lentiviral production for CRISPR/Cas9 or CRISPR interference**. gRNA sequences targeting rodent Esrra or Esrrg were designed by CRISPR gRNA Design tool in ATUM (https://www.atum.bio/eCommerce/cas9/input). All DNA oligonucleotides used for cloning were generated by GENEWIZ, LLC. (South Plainfield, NJ) and listed in Supplementary Table 5. The designed gRNA was cloned into lentiCRISPRv2 hygro[68] (a gift from Dr. Brett Stringer, the University of Queensland; Addgene plasmid # 98291) or lentiCRISPRv2 blast[69] (a gift from Dr. Brett Stringer, the University of Queensland; Addgene plasmid # 98293). Phosphorylated and annealed DNA oligonucleotides were cloned into Esp3I (ThermoFisher, FD0454)-digested lentiCRISPRv2[63]. For knocking down expression of PPARGC1A in hiPSC-CMs with CRISPR interference (CRISPRi), pLV hU6-sgRNA hUbC-dCas9-KRAB-T2a-Puro[70] (a gift from Dr. Charles Gersbach; Addgene plasmid # 71236) was employed. plentiCRISPRv2 or pLV hU6-sgRNA hUbC-dCas9-KRAB-T2a-Puro with cloned gRNA for each target was transfected to 90% confluent 293 FT cells (Thermo Fisher Scientific, R70007) in T-25 flask with lipofectamine 2000 (ThermoFisher, 11668019) according to manufacturer's protocol with psPAX2 (a gift from Dr. Didier Trono; Addgene plasmid # 12260) and pMD2.G (a gift from Dr. Didier Trono; Addgene plasmid #12259). The control lentivirus for CRISPRi did not express gRNA. Eighteen hours after the transfection, 6 mL fresh 30% FBS-supplemented DMEM with L-glutamine, 4.5 g/L glucose and sodium pyruvate (Cellgro, 10-013-CM), called harvesting media, was supplied for the transfected cells. The media containing lentiviruses were harvested 2, 3, and 4 days later and fresh harvesting media was supplied each harvesting time. After the pooled media was briefly centrifuged to spin down the cells, the lentivirus in the supernatant was concentrated by around 10 times with Lenti-X™ Concentrator (Takara, 631232) according to the manufacturer's instruction.

**Luciferase assay**. pG5luc[71] was employed to monitor the activity of the ERRα or γ-Gal4-fused proteins. One hundred seventy-five nanograms of a luciferase reporter was cotransfected with the indicated expression plasmids along with 5 ng of CMV promoter-driven Renilla luciferase (Promega) to control for transfection efficiency[66] per one well of a 24 well plate. For pG5luc, 70 ng of PGC-1α, GATA4 or GATA6, and 16.6 ng of ERRα-Gal4 or ERRγ-Gal4 plasmids were used. For SMYD1-luc, 70 ng of PGC-1α and 16.6 ng of ERRα or ERRγ plasmids were used. For TNNI3, COX6A2, or MYH6-7 enhancer-luc, 70 ng of expression plasmid was used. H9c2 myoblast (ATCC, CRL-1446) and AD-293 cells were cultured at 37 °C and 5% CO₂ in 10% FBS-supplemented DMEM with L-glutamine, 4.5 g/L glucose and sodium pyruvate supplemented (Cellgro, 10-013-CM). AC16 cells (Sigma-Aldrich, SCC109)[72] were cultured at 37 °C and 5% CO₂ in 12.5% FBS- and 2 mM L-glutamine-supplemented (ThermoFisher, 25030-081)-DMEM/Nutrient Mixture F-12 Ham (Sigma-Aldrich, D6434-500ML). Transient transfection of the indicated luciferase reporters, expression vectors, and Renilla control plasmid (Promega) was performed using Lipofectamine 2000 (ThermoFisher, 11668019) for AD-293 cells or Lipofectamine 3000 (ThermoFisher, L3000015) for H9c2 or AC16 cells as per the manufacturer's protocol. pcDNA3.1-EGFP was used as a control, and the

same amount of DNA was transfected to all tested groups. 48 h after the transfection, luciferase assay was performed using Dual-Glo Luciferase Assay System (Promega, E2920) as per the manufacturer's recommendations. The luciferase activity was detected with SYNERGY H1 (BioTeK, 11-120-533) and BioTek Gen5 3.03 software.

**Immunoprecipitation**. Twenty four hours after plating AD-293 on 150 mm culture dish (Greiner, 639160) at 6–7 × 10⁶ cells per dish, pcDNA3.1-GATA4-HA, 3xFLAG-ERRγ, and PGC-1a-Myc/His[66] were transfected to AD-293 with lipofectamine 2000 (ThermoFisher, 11668019) per manufacturer's recommendations. Alternatively, FLAG-tagged ERRγ was overexpressed with adenovirus system at MOI100 in the confluent α-Skin hiPSC-CMs on 10 cm culture dish. 48 h after the gene transfection, the cells were incubated with 1 mL of nuclear isolation buffer [10 mM HEPES (pH7.9), 1.5 mM MgCl₂, 10 mM KCl, 340 mM sucrose, 10% glycero] freshly supplemented with cOmplete EDTA-free (Roche, 11873580001) for 3 min on ice, and the lysate was further incubated with 0.1% Triton X-100 for 3 min on ice. Nuclei pellet was obtained with centrifugation at 1500 × g for 5 min at 4 °C. After centrifugation at 1500 × g for 5 min at 4 °C, nuclei pellet was suspended with 500 μL of lysis buffer [10 mM HEPES (pH7.9), 3 mM MgCl₂, 5 mM KCl, 150 mM NaCl, 0.5% NP-40, benzonase (Sigma, E1014-25KU)] and rotated for 45 min at 4 °C. Clear supernatant was centrifuged at 21,130 × g for 20 min at 4 °C. Fifty microliters of lysate was saved as input. The rest of lysate was mixed with 50 μL Anti-DYKDDDDK Magnetic Agarose (Thermo Scientific, A36797) or 25 μL Anti-HA Magnetic Beads (Thermo Scientific, 88836) or 25 μL Anti-c-Myc Magnetic Beads (Thermo Scientific, 88842). IgG-conjugated Dynabeads protein G were prepared with 1 μg mouse control IgG (Proteintech, B900620). The mixture of lysate and antibody-conjugated magnetic beads were rotated for 30 min for AD-293 or for 90 min for hiPSC-CMs at 4 °C. The magnetic beads were washed four times with lysis buffer. Beads were boiled for 5 min at 95 °C in 35 μL of 2× Laemmli sample buffer (BIO-RAD, #161-0737).

**Fatty acid oxidation assay**. FAO assay using [³H]-palmitic acid was performed[73] with hiPSC-CMs at approximately D20 plated in a Matrigel-coated 24-well plate. Adenoviral ERRγ or GFP OE was performed as described in the hiPSC culture system section. Following ERRγ or GFP OE, cells were rinsed three times with PBS and then incubated in 125 μM [³H]-palmitic acid (PerkinElmer, NET043001MC, 60 Ci/mmol) bound to fatty acid-free albumin containing 1 mM carnitine for 2 h at 37 °C. The cell medium was transferred to a tube containing cold 10% trichloroacetic acid. The tubes were centrifuged at 8500 × g for 10 min at 4 °C. The supernatant was immediately removed, mixed with 6 N NaOH, and applied to ion-exchange resin (DOWEX 1; Sigma-Aldrich). The eluate was collected, measured by liquid scintillation analyzer (PerkinElmer) and normalized to total protein amount. The amount of cell protein was measured by Micro BCA protein assay kit (Thermo Scientific, 23235).

**LC/MS/MS quantitation of metabolites**. Metabolomics was performed by the Metabolomics Core at Sanford Burnham Prebys Medical Discovery Institute in Orlando, FL. Metabolomics analyses were performed on extracts from hiPSC-CMs following adenoviral OE GFP or ERRγ, 2 days after the viral infection. BSA-conjugated palmitic acid at 100 μM with 1 mM carnitine was supplemented in the media for the acylcarnitine metabolomics assay. The cells were immediately washed by cold PBS and snap frozen in liquid nitrogen. A volume of acetonitrile/0.6% formic acid, equivalent to the volume of each lysate, was added to the first set of samples for extraction of organic acids. A 50 μL aliquot of the homogenate was used for the organic acid assay. The second set of frozen cell slush samples was similarly extracted like the organic acids, and a 100 μL aliquot of the homogenate was used for the acylcarnitine assay.

For protein determinations, 10 μL aliquots of each thawed cell slush were removed and immediately frozen. Protein concentrations of the cell slush where determined using the Pierce BCA Protein Assay Kit, with BSA as the standard.

To extract organic acids, aliquots of cell homogenates were spiked with isotopically-labeled internal standards, extracted in ethylacetate, and centrifuged[74]. Then, 50 μL of ethylacetate extracts were dried, and organic acids were derivatized with O-benzylhydroxylamine using 1-ethyl-3-(3-dimethylaminopropyl) carbodiimide (EDC) coupling chemistry according to prior studies[74]. Derivatized organic acids were quantitated using multiple reaction monitoring on a Dionex UltiMate 3000 HPLC/Thermo Scientific Quantiva triple quadrupole mass spectrometer[74].

Acylcarnitines were extracted from aliquots of cell homogenates by spiking with isotopically labeled internal standards, addition of 800 μL of ice-cold methanol, and centrifuging to pellet precipitated protein[75]. Then, 100 μL of the methanolic extract was dried down, and acylcarnitines were derivatized using the EDC coupling chemistry above. Derivatized acylcarnitines were quantitated using multiple reaction monitoring on an Agilent 1290 HPLC/6490 triple quadrupole mass spectrometer[75]. All procedures were performed by the Metabolomics Core at Sanford Burnham Prebys Medical Discovery Institute in Orlando, FL.

**Immunoblot analysis and antibodies**. Whole-tissue lysates or nuclear protein lysates were subjected to SDS-PAGE and transferred to a nitrocellulose

membrane[10]. The binding of primary antibodies was detected by IR Dye 800CW Donkey anti-Rabbit IgG (LICOR, 926-32213), IR Dye 800CW Donkey anti-Mouse IgG (LICOR, 926-32212), IR Dye 680RD Donkey anti-Mouse IgG (LICOR, 926-68072), or IR Dye 680RD Donkey anti-Rabbit IgG (LICOR, 926-68073) at dilutions of 1:15,000 and scanned with LI-COR Odyssey infrared imaging system or Odyssey Fc (LI-COR Biosciences). Immunoprecipitated samples were detected with Veri-Blot for IP Detection Reagent (HRP) (Abcam, ab131366, 1:10,000 dilution) and SuperSignal™ West Duration Substrate (ThermoFisher, 34075). Odyssey Fc. REVERT™ Total Protein Stain Kits (LI-COR Biosciences) were used to stain the whole protein on the western blotting membrane. The following antibodies were used: ERRα; SMYD1; VDAC; ACSL3; TNNI3 (Abcam catalog and clone #s: ab76228; EPR46Y, ab181372; EPR13574(B)-30, ab15895, ab151959, and ab47003 respectively); His tag; α-tubulin; ACSL1 (Cell Signaling Technology catalog and clone #s: 12698S; D3I1O, 3873; DM1A, and 9189; D2H5 respectively); HA tag and MYL2 (Proteintech catalog #s 51064-2-AP and 10906-1-AP); α-Actinin, FLAG M2, TNNI1, ACTB, and PGC-1α (Sigma-Aldrich catalog and clone #s: A7732-100UL; EA-53, F1804-50UG; M2, AV42117-100UL, A5316; AC-74, and ST1202; 4C1.3); GATA4 and MYBPC3 (SANTA CRUZ BIOTECHNOLOGY catalog and clone #s, sc-25310; G-4, and sc-137180; E-7). The anti-ERRγ for immunoblotting endogenous ERRγ was originally raised by Dr. Ronald Evans (Salk Institute) and validated by us and others[10–12,76,77]. For detecting ERRγ in IP samples, anti-ERRγ generated by the collaboration of the Daniel Kelly and Anastasia Kralli laboratories[10] was used. All primary antibodies were diluted to 1:1,000 except for anti-GATA4 and MYBPC3 (1:200). Protein quantification was performed using LICOR Image Studio Version 5.2. (LI-COR Biosciences) and normalized to an internal control protein such as α-Tubulin or total protein levels.

**RNA isolation and quantitative RT-PCR.** Total RNA was isolated using the RNeasy Mini Kit (QIAGEN, 74104) or miRNeasy Mini Kit (QIAGEN, 217004) according to the manufacturer's instructions. cDNA was synthesized using the Affinity Script cDNA Synthesis Kit (Stratagene, 600559) with 0.5 μg total RNA. PCR reactions were performed in triplicate (QuantStudio 6 Flex, applied biosystems) with specific primers for each gene. QuantStudio Real-Time PCR Software v1.7 was utilized. Primer sets are listed in the Supplementary Table 4. The expression of human *RPLP0* (also known as 36B4) was used respectively to normalize all gene expression data. The following thermal cycler conditions were used: preincubation at 95 °C for 3 min, amplification at 95 °C for 10 s, and at 60 °C for 20 s. ΔΔCt method was used for calculating the gene expression levels.

**RNA-seq library preparation/sequencing.** For RNA-seq with ERRα/γ KO hiPSC-CMs, RNA library preparations and sequencing reactions were conducted at GENEWIZ, LLC. (South Plainfield, NJ). RNA samples received were quantified using Qubit 2.0 Fluorometer (Life Technologies) and RNA integrity was checked using Agilent TapeStation 4200 (Agilent Technologies). RNA-sequencing libraries were prepared using the NEBNext Ultra RNA Library Prep Kit for Illumina using the manufacturer's instructions (NEB). Briefly, mRNAs were initially enriched with Oligod(T) beads. Enriched mRNAs were fragmented for 15 min at 94 °C. First-strand and second-strand cDNA were subsequently synthesized. cDNA fragments were end repaired and adenylated at 3'ends, and universal adapters were ligated to cDNA fragments, followed by index addition and library enrichment by PCR with limited cycles. The sequencing library was validated on the Agilent TapeStation (Agilent Technologies), and quantified by using Qubit 2.0 Fluorometer (Invitrogen) as well as by quantitative PCR (KAPA Biosystems).

The sequencing libraries were clustered on a single lane of a flowcell. After clustering, the flowcell was loaded on the Illumina NovaSeq 6000 according to manufacturer's instructions. The samples were sequenced using a 2 × 150 bp paired-end (PE) configuration. Image analysis and base calling were conducted by the HiSeq Control Software. Raw sequence data (.bcl files) generated from Illumina HiSeq was converted into fastq files and de-multiplexed using Illumina's bcl2fastq 2.17 software. One mismatch was allowed for index sequence identification.

For RNA-seq on hiPSC-CMs overexpressed with ERRγ, the library preparation and sequencing reaction was performed at the Analytical Genomics Core at Sanford Burnham Prebys Medical Discovery Institute. The total RNA was subjected to ribodepletion to remove rRNA. The remaining RNA was purified, fragmented, reverse transcribed, adapter ligated and finally PCR amplified to enrich the DNA fragments. These mRNA-seq libraries were pooled per lane of PE 50 bp sequencing on an Illumina HiSeq 2500 instrument. Illumina's Basespace was used to convert bcl files to FASTQ files and de-multiplex the samples.

**RNA-seq analysis.** Transcript levels were quantified with Salmon[78] using human reference genome GRCh38. Transcript data were aggregated to produce gene-level quantifications, genes with less than 10 reads across all samples were removed, and tests for differential expression were performed using DESeq2[79]. We considered significantly regulated genes with Benjamini-Hochberg FDR < 0.05 and fold change at least 1.5 in either direction for further analyses. Two (ERRα/γ KO1) or three (WT Control, ERRα/γ KO6, GFP OE, or ERRγ OE) biological replicates were used for the indicated analyses. The correlation between biological replicates of the same group in all RNA-seq datasets newly generated in this study is above 0.98. The heatmaps in Fig. 1c, Supplementary Fig. 2g, and Supplementary Fig. 8a were

generated with MORPHEUS (https://software.broadinstitute.org/morpheus/), and Fig. 3d was generated with the custom code in R (https://github.com/batmanovkn/err_gata4_cardiomyocytes).

**Intersection analysis with RNA-seq datasets.** The significantly downregulated genes (Benjamini–Hochberg FDR < 0.05, FC < −1.5, commonly regulated in both KO lines) in ERRα/γ KO hiPSC-CMs at D22 were intersected with the reported downregulated genes in G296S GATA4 hiPSC-CMs at D32[19].

Genes significantly downregulated (Benjamini-Hochberg FDR < 0.05, FC < −1.5) in ERRα/γ KO hiPSC-CMs (derived from both KO1 and 6) were intersected with significantly downregulated (Benjamini-Hochberg FDR < 0.05) genes in right ventricular samples from HFrEF patients (https://zenodo.org/record/4114617#.YWrnTNnMJ0w). GO term enrichment analyses with the indicated gene sets were performed with g:Profiler[80]. Statistical significance of gene set overlaps was assessed with a Fisher's exact test using all Ensembl genes as background.

**Intersection analysis with RNA-seq and ChIP-seq datasets.** To identify ERR-activated or -suppressed targets, the genes closest to ERRγ peaks, as determined by Homer's annotatePeaks.pl, where the peak is within 5 kbp from the transcriptional start site (TSS) of the gene, were intersected with the significantly (Benjamini-Hochberg FDR < 0.05, |FC| > 1.5) downregulated- or upregulated genes in both lines (KO1 and KO6) of ERRα/γ KO hiPSC-CMs.

**ChIP-qPCR and ChIP-seq.** ChIP was performed with the chromatin samples isolated from D22 hiPSC-CMs[10]. The cells were lysed with the ChIP lysis buffer (1% Triton x-100, 0.1% SDS, 150 mM NaCl, 1 mM EDTA, and 20 mM Tris, pH 8.0) after crosslinking with 1% paraformaldehyde (diluted Pierce™ 16% Formaldehyde (w/v), 28908 with PBS) and Disuccinimidyl glutarate (ProteoChem, c1104-1gm). After quenching the crosslinking with 1.25 mM Glycine, the isolated chromatin samples were sonicated with Bioruptor (Diagenode) with the low setting (30 sec on; 30 s off for 10 min × 2). Acetylated histones, ERRγ or overexpressed GATA4-HA was immunoprecipitated using 5 μg anti-H3K27ac (Abcam, ab4729) or anti-ERRγ[10]-conjugated Dynabeads protein G (Thermo Scientific, 10004-D) or 25 μL anti-HA-conjugated magnetic beads (Thermo Scientific, 88836) overnight at 4 °C respectively. Chromatin-antibody complexes were washed with IP Wash Buffer 1 (1% Triton, 0.1% SDS, 150 mM NaCl, 1 mM EDTA, 20 mM Tris, pH) once, IP Wash Buffer 2 (1% Triton, 0.1% SDS, 500 mM NaCl, 1 mM EDTA, 20 mM Tris, pH 8.0, and 0.1% Sodium Deoxycholate) twice, IP Wash Buffer 3 (0.25 M LiCl, 0.5% NP-40, 1 mM EDTA, 20 mM Tris, pH 8.0, 0.5% Sodium Deoxycholate) once, and IP Wash Buffer 4 (10 mM EDTA and 200 mM Tris, pH 8) once. The washed chromatin-antibody complexes were eluted from Dynabeads with Bead Elution Buffer (1% SDS and 100 mM NaHCO₃). The eluted samples were incubated with 200 mM NaCl at 65 °C overnight. After incubating the samples at 37 °C for 30 min with 20 μg RNase, they were incubated with 40 mM Tris-HCl, 10 mM EDTA, and 40 μg Proteinase K at 45 °C for 2 h. DNA fragments were purified using a QIAquick PCR Purification Kit (QIAGEN). Barcoded ChIP-seq libraries were made from total 2 μg ChIP DNA using Ovation Ultralow Library Systems (Nugen) from two biological replicates according to the manufacturer's instructions. Eight ChIP-seq libraries were pooled per lane of single-end 50 bp sequencing on an Illumina HiSeq 2500 instrument by the Analytical Genomics Core at Sanford Burnham Prebys Medical Discovery Institute. All ChIP-qPCR primers are listed in Supplementary Table 4. Stem cell enhancer region on POU5F1 was used as a negative control in ChIP-qPCR experiments.

**ChIP-seq analysis.** Reads were first aligned to reference genome GRCh38 using Bowtie2[81] with parameters -N 1, after which the duplicates were removed and only uniquely mapped reads selected using samtools. Peaks were called on pooled replicate datasets with Homer's findPeaks[82], using corresponding KO (for ERRγ) or input (for H3K27ac) experiments as controls, with default parameters. For motif analysis, we used Homer's findMotifsGenome.pl with the parameters: hg38 -size 200 -S 15 -len 10,12,14,16,18. We have used the default background generated by the Homer package, which is genomic regions that match the GC-content distribution of the input sequences unless stated otherwise. Genes nearest to peaks were found with Homer's annotatePeaks.pl. The independent biological replicate for H3K27ac ChIP-seq in each WT and ERRγ KO was highly correlated (WT r = 0.99, ERRγ KO r = 1.00). In addition, we confirmed that average H3K27ac ChIP-seq signals around TSS of all genes has the expected shape in both genotypes, indicating the quality of this ChIP-seq study is high as shown in Supplementary Fig. 3c.

**Intersection analysis with ChIP-seq datasets.** We used tools from Homer to quantify H3K27ac reads inside ERRγ peaks and to intersect the peak regions. Homer was used to compute statistical significance of region overlaps with the Fisher's exact test, using effective genome size 9·10⁸ base pairs. Peaks near genes were selected as those located within 5 kb from gene's TSS. The motif search on ERRγ and H3K27ac overlapped peaks around ERR-activated or -suppressed targets was performed with Homer's findMotifsGenome.pl with the parameters, hg38 -size 200 -S 15 -len 10,12,14,16,18, with modified background, which was ERRγ and H3K27ac overlapped regions not associated with differentially regulated genes in ERRα/γ KO hiPSC-CMs. The intersection analysis to define ERRγ + GATA4 target

regions was conducted with GATA4 peaks in the published ChIP-seq [GSE85631] and ERRγ peaks in our ERRγ ChIP-seq [GSE113784] previously published[10]. GATA6 ChIP-seq information was previously published[18]. To define SE + ERRγ targets, the intersection analysis was conducted with ERRγ peaks and SEs previously determined with MED1 ChIP-seq [GSE85631] in hiPSC-CMs[19]. The overlapped regions between the ERRγ peaks and indicated ChIP-seq datasets were defined as those merged peaks of ERRγ peaks and these datasets found by Homer. For intersections with ERRγ and GATA4 or GATA6 peaks, the maximum distance between peaks to be considered intersecting was set to 400 (-d 400 option of mergePeaks). For SEs, only the simple overlap was considered (-d given). To determine statistical significance of peak overlaps, Fisher's exact test was used with genome size $9 \cdot 10^8$. Each ChIP-seq peak around representative targets was visualized with Integrative Genomics Viewer 2.4.18[83]. To calculate the distance distribution between ERRγ and GATA4 within the same SEs, we found the nearest ERRγ and GATA4 peaks in the 99 SEs where both ERRγ and GATA4 peaks are found. The space between the nearest peaks was taken as the representative distance in the SE. If the nearest ERRγ and GATA4 peaks overlapped, the distance was taken to be 0. We performed differential motif enrichment scans between ERRγ + GATA4 and ERRγ-GATA4 with Homer's findMotifsGenome.pl, specifying the other peak set as custom background.

**Analysis with Cistrome Database Browser**. Toolkit for Cistrome Database Browser (http://dbtoolkit.cistrome.org/) was employed to find most overlapped cistrome datasets with all peaks in each ERRγ-GATA4, GATA4-ERRγ, and ERRγ + GATA4 dataset. Transcriptional regulators which are not significantly expressed in hiPSC-CMs (FPKM < 0.1 in our RNA-seq from WT control hiPSC-CMs) were removed from the results. Statistical significance of the identified overlaps was assessed using Fisher's exact test with genome size $9 \cdot 10^8$, applying Bonferroni correction for multiple tests ($N = 11,348$, number of Human Factor datasets in CistromeDB).

**ATAC-seq library preparation**. Two biological replicates of WT and ERRα/γ KO hiPSC-CMs from distinct two lines (KO1 and KO6) around D20 were digested with Accutase (Sigma-Aldrich, A6964-500ML) and 100,000 cells were collected with centrifugation for 15 min at $500 \times g$ in 4 °C[84,85]. The cell pellet in each sample was resuspended with 50 μL cold lysis buffer (10 mM Tris-HCl, 10 mM NaCl, 3 mM MgCl$_2$, 0.1% IGEPAL CA-630). The nuclei were pelleted by centrifugation for 30 min at $500 \times g$ in 4 °C. Supernatants were discarded and nuclei were resuspended in 50 μl reaction buffer [2.5 μl of Tn5 transposase and 25 μl of TD buffer from a Nextera DNA Library Prep Kit (llumina, FC-121-1030) and 22.5 μl nuclease-free H$_2$O]. The reaction was incubated at 37 °C for 30 min, and subsequently the reaction mixture was purified using MinElute PCR Purification Kit (Qiagen, 28004). The purified transposed DNA was amplified with NEBNext High-Fidelity 2 X PCR Master Mix (New England Biolabs, M0541S) with Nextera Index Kit (Illumina, FC-121-1011). The amplified DNA samples were purified with AMPure XP (BECKMAN COULTER, A63881). DNA concentration was measured with a Qubit Fluorometer (Thermo Fisher Scientific) and library sizes were determined using Agilent High Sensitivity DNA kit (Agilent, 5067-4626) on 2100 Bioanalyzer (Agilent, G2939BA). The ATAC-seq libraries were sequenced with a Novaseq sequencer (Illumina) with pair end 150 bp, and the sequencing quality control was performed by Next-Generation Sequencing Core at the Perelman School of Medicine, University of Pennsylvania.

**ATAC-seq analysis**. Reads were aligned to GRCh38 with Bowtie2. We used Homer to identify open ATAC regions in the pooled WT samples. Regions which have differential ATAC-Seq signal between pooled WT and ERRα/γ KO were called using Homer/DESeq2, selecting those with Benjamini-Hochberg FDR < 0.05 and fold change at least 2 in any direction. The ATAC-Seq datasets meet the high-quality standards set by ENCODE in the majority of key measures including signal level (Supplementary Fig. 4a), fragment length distribution expected of ATAC signals (Supplementary Fig. 4b), and reproducibility (Supplementary Fig. 4c). IDR values were calculated using Python package idr[86]. For motif analysis, we used Homer's findMotifsGenome.pl with the parameters: hg38 -size 200 -S 15 -len 10,12,14,16,18. We have used the default background generated by the Homer package, which is genomic regions that match the GC-content distribution of the input sequences. ATAC-seq peaks and the indicated ChIP-seq peaks were intersected with Homer's mergePeaks program. To determine statistical significance of peak overlaps, Fisher's exact test was used with genome size $9 \cdot 10^8$.

**Enrichment analysis**. Enrichment analyses were performed with g:Profiler[80] web service. Significantly enriched GO or KEGG pathway terms (Benjamini-Hochberg FDR < 0.05 and enrichment FC > 2) were presented as bar graphs. 8,893 genes fairly expressed in hiPSC-CMs (FPKM ≥ 10) in our RNA-seq data were used as a background for all enrichment analyses in this study.

**Statistics and reproducibility**. For two group comparisons, a two-tailed student's t-test was performed. For multiple comparisons, 1 or 2-way ANOVA with Dunnett's post-hoc test or with Tukey's post-hoc test was employed to determine

statistical significance as indicated in the figure legends. Significant differences were defined as a p value < 0.05. Exact p values for each bar graph are provided in Source Data, and for Venn diagram and motif analyses, they are presented in the figures. The study did not additionally correct p-values for multiple testing across experiments in the whole study. The ROUT method was used to identify outliers. Microsoft Excel version 16 and GraphPad Prism 8 or 9 software were used for graphing and statistical analysis. All bars represent the means ± SEM. n denotes biological replicates. The representative immunoblot images were replicated with at least two biological replicates over two independent experiments.

**Reporting summary**. Further information on research design is available in the Nature Research Reporting Summary linked to this article.

## Data availability

GRCh38 (http://ftp.ensembl.org/pub/release-99/fasta/homo_sapiens/) was used as reference genome. ENSEMBL gene annotations v99 were used (http://ftp.ensembl.org/pub/release-99/gtf/homo_sapiens/). The RNA-seq, ChIP-seq, and ATAC-seq data generated in this study have been deposited in NCBI's Gene Expression Omnibus (GEO) under series accession number GSE166064. ERRγ ChIP-seq data is available in NCBI's GEO with GSE113784. GATA4 and MED1 ChIP-seq data that support the findings of this study are available in NCBI's GEO with GSE85631. Significantly regulated genes in G296S GATA4 hiPSC-CMs were obtained in https://doi.org/10.1016/j.cell.2016.11.033. GATA6 ChIP-seq information was obtained from https://elifesciences.org/articles/53278/figures#content[18]. The HFrEF RNA-seq data[49] was obtained from online data repository (https://zenodo.org/record/4114617#.YWrnTNnMJ0w). Source data are provided with this paper. Uncropped immunoblot images are provided in the Source Data. Source data are provided with this paper.

## Code availability

Code used to analyze the datasets in this paper is available at GitHub (https://github.com/batmanovkn/err_gata4_cardiomyocytes).

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

## Acknowledgements

We thank Teresa C. Leone for critical review of the manuscript, and manuscript preparation. We are grateful to Dr. Anastasia Kralli (Johns Hopkins Medical School) for our collaboration to generate the ERRγ antibody; Dr. Huei-Sheng Vincent Chen (Indiana University School of Medicine) for providing α-Skin hiPSC; and Dr. Deepak Srivastava (Gladstone Institutes) for providing WTC11 hiPSC. We acknowledge the following Core Facilities at Sanford Burnham Prebys Medical Discovery Institute at Lake Nona (SBP-LN): Analytical Genomics Core and the Bioinformatics Core who performed the RNA-seq and ChIP-seq, and the Metabolomics Core. We extend special thanks to the Next-Generation Sequencing Core and the Metabolomics Core of the Perelman School of Medicine at the University of Pennsylvania. This work was supported by NIH R01 HL058493 (D.P.K.) and a postdoctoral fellowship from the American Heart Association #14POST20490309 (T.S.).

## Author contributions

T.S. and D.P.K. designed, analyzed, and wrote the manuscript. T.S. conducted all experiments, with assistance from L.L. and Y.G. R.B.V., L.L., and D.P.K. participated in scientific discussions. K.B. and S.W. conducted bioinformatics analyses. Y.G. performed several immunoprecipitation experiments and provided advice for conducting immunoprecipitation. All authors edited and approved the manuscript.

## Competing interests

The authors declare no competing interests.
