## [Peer Review File · Nature Communications]

The Nuclear Receptor ERR Cooperates with the Cardiogenic Factor GATA4 to Orchestrate Cardiomyocyte MaturationREVIEWER COMMENTS

Reviewer #1 (Remarks to the Author):

This is a well-written manuscript that describes exciting and important findings. The authors primarily overlaid and integrated large next generation sequencing data sets in order to uncover mechanisms whereby ERRgamma cooperates with GATA4 and PGC1-alpha to control cardiomyocyte structural and metabolic gene programs, respectively. The approaches, methods and data are, for the most part, explicitly explained and accessible. This manuscript is extensive and the data are convincing. The work seems to be ideally suited to and within the scope of Nature Communications. I have only minor suggestions for improvement.

Minor comments:

1. The description of Fig. 1a in the text (Page 5) describes transcripts, whereas the data includes protein levels of ERRs. This should be changed to say 'Transcripts encoding the ERR family of nuclear receptors as well as protein levels of ERRalpha and gamma...'
2. Page 4 'Cistromic analyses in human induced pluripotent stem cell-derived cardiomyocytes (hiPSC-CMs) suggested that ERR γ is a direct activator of adult contractile and ion channel genes in addition to regulating canonical target genes involved in mitochondrial energetic processes such as FAO and oxidative phosphorylation (OXPHOS).' Please provide references for this statement.
3. In the Discussion, there should be mention of the limitations of hiPSC-CMs as a model of chronic cardiac conditions such as heart failure. In particular, with regard to immature sarcomere structure, seeing as this is one of the key processes claimed to be regulated by ERR γ -PGC1 α -GATA4 cooperation.
4. Following on from the previous point, whilst reference is made to enrichment of binding motifs for GATA, MEF2 and ERRs in human embryonic hearts, the potential lack of translational relevance should also be noted. Evaluation of RNA-seq data from hearts of human CHD patients, such as in doi.org/10.1038/ncomms12824, would improve the manuscript.
5. The sex of the source of the hiPSCs should be stated in the Methods section.
6. For RNA-seq data, any minimal threshold reads/RPM applied in a minimal number of samples should be stated.
7. I believe a methods section on Gene Ontology analysis, including the programs used and statistical analysis and thresholds applied, is missing.
8. The method used for multiple comparison testing to calculate FDR adjusted p-values in NGS data sets used should be stated (Bonferroni, Benjamini-Hochberg..?).
9. As I understand it, Fig. 3C is reflecting data in Table S1. I think the clusters and/or z-score values should then be indicated in Table S1. A reference to which Suppl Tables data figures are included in figure legends would help orientate the reader.

Reviewer #2 (Remarks to the Author):

Re: Sakamoto et al., The Nuclear Receptor ERRg Cooperates with the Cardiogenic Factor GATA4 to Orchestrate Cardiomyocyte Differentiation

The authors previously reported that ERRa and ERRg, known to regulate mitochondrial gene expression in multiple tissues, also regulates postnatal maturation of cardiomyocytes, including expression of cardiomyocyte-specific genes in addition to mitochondrial genes. This raised the question of how these broadly deployed transcription factors could drive cardiomyocyte specific gene programs. The authors pursued this question in this study, using human iPSC-CMs as the primary model system. The authors found that ERRg interacts with GATA4, an important cardiac transcription factor. They demonstrate that ERRg and GATA4 are present in the same protein complexes, and that GATA4 potentiates ERRg transcriptional activity. ERRg-activated genes that were co-bound by GATA4 were enriched for mature cardiomyocyte specific genes, such as genes important for contraction, Ca²⁺ handling, and electrical activity, whereas those that were not co-bound by GATA4 were enriched for mitochondrial metabolism. Both gene classes were activated by the ERRg co-activator PGC1a.

Overall the main advances of this study are the demonstration that ERR activates cardiomyocyte specific genes and not just mitochondrial genes, and that it does through collaboration with GATA4. The general concept of general transcription factors adopting tissue specific activity through interaction with tissue-specific transcription factors is certainly not novel. Nevertheless this new information on ERR is interesting and valuable.

The manuscript had a number of weaknesses that decreased enthusiasm, as described below.

Major points:

1. Data quality and analysis.

- The GEO record shows that in most cases either duplicate or triplicate data were obtained. QC analyses of the data, such as correlation between replicates, was not presented.
- Information about how replicates were combined to define consensus peaks and signals is absent.
- It would be helpful to provide more information about each dataset. For example, a volcano plot of RNA-seq data showing number of up and down regulated genes, and their definitions.
- ChIP-seq and ATAC-seq: how many peaks were there? These numbers could be shown in Fig. S3, Fig. 3. The number of ATAC-seq peaks appears to be low, leading to questions about data quality. Does the data meet ENCODE guidelines for ATAC-seq data?
- When discussing overlaps between region sets, the statistical significance of the overlap should be included (e.g. permutation test or hypergeometric test).
- For motif analysis and GO analysis, the appropriate background gene sets or regions should be used. For instance, if hiPSC-CM superenhancers have GO terms for cardiac muscle cell differentiation or myofibril assembly, is it surprising that the ERRg-GATA4 subset is also? What is the enrichment found in this subset when the remaining SEs are used as the background? Similar considerations for other GO and motif analyses. As another example, is there differential motif enrichment in regions with decreased H3K27ac in ERR KO, compared to regions with no change in H3K27ac? Specifically is GATA4 motif enriched in all ERRg regions, or only in the ERRg subset with reduced H3K27ac?

2. Both ERR and GATA transcription factor families are redundant – ERR knockout requires ablation of both ERRa and ERRg, and GATA4 ablation requires ablation of both GATA4 and GATA6. This issue is not well addressed in this study, which focuses on ERRg-GATA4 interaction. Given the redundancy, one would expect that ERRg-GATA6, ERRa-GATA4, and ERRa-GATA6 interactions should also potentiate activation of the same sets of genes.

3. The manuscript demonstrates that over-expressed ERRg and GATA4 co-precipitate. However, this should be demonstrated at the level of endogenous proteins, and appropriate controls to demonstrate IP specificity should be included. Evidence of direct ERRg-GATA4 interaction is lacking.

GATA4 was shown to potentiate transactivation by ERRg in an assay that did not depend upon GATA4 binding activity. However, the motif enrichment and genome browser shots showing GATA4 and ERRg binding to adjacent regions suggests that GATA4 DNA binding activity is also involved. Whether or not GATA4 and ERRg influence each other's DNA binding activity and chromatin occupancy was not investigated. Indeed, "cobinding" in the current analysis is defined as both

factors binding to the same gene, and whether there is significant co-binding to the same enhancers is not addressed.

It is also notable that the two GATA4 mutants that significantly affected ERRg transcriptional activity both impair DNA binding. Could GATA4 DNA binding be influencing the activity of the GAL4 reporter?

Since SEs are large, co-binding of ERRg and GATA4 to the same SE does not provide information about potential cooperative binding. What is the distance between ERRg and GATA4 sites in SEs?

4. "Enhancers" are usually considered distal regions, distinct from "Promoters". However, this distinction is not made in this manuscript. For example, in Fig. 3b the selected examples are all promoters.

5. The manuscript tends to jump into the analysis of specific examples without providing a more global view. To what extent do the conclusions made for these cherry-picked examples hold in a more unbiased, genome-wide view?

6. The manuscript argues that ERRg + GATA4 is important for the expression of maturation-related cardiomyocyte structural genes, with TNNI3 used as the example. The hiPSC-CM system is not ideal for studying maturation because maturation is arrested in this system. The authors have access to the relevant knockout mice. Can they provide data from the mouse system to demonstrate that ERR + GATA4 is required for maturation. Is there evidence that GATA4 is required for CM maturation?

7. What features distinguish ERRg peaks that are dependent or independent of GATA4? In addition to differential motif analysis, the authors could mine existing ChIP-seq data from hiPSC-CMs to define features or TF occupancy that differs between these ERRg peak classes.

7. Fig. 1.

a-b, normalization of all the data to the value at day 0 can lead to some confusion. For example, ERRa has no significant change during the time course, but it is expressed as robustly as ERRg. Can the data be shown as expression relative to house keeping gene at each time point without normalizing day 0 to 1.

d, please indicate statistical significance.

8. Fig. 2.

b. As controls, plot the H3K27ac density at non-ERRg peaks.

H3K27ac – ERRg overlapping regions can be classified into those associated with ERR-activated, ERR-suppressed, and ERR-unchanged genes. Motif analysis and H3K27ac plots could be made for these three classes.

9. Fig. 4

e. What accounts for the increased width of ATAC peaks in KO? Please include other regions (e.g. H3K27ac+, ERRg-) as control regions to show the reduced ATAC signal is specific to ERRg regions.

10. Fig. 7

e. The two GATA4 point mutants that reduced activity of the reporter both are defective in DNA binding. I.

Minor

1. ERRg-deficiency impacted many fetal to adult isoform switches as demonstrated by marked reduction in expression of TNNI3 – only example provided is TNNI3. What are the other switches affected referred to by "many".

2. ...downregulated by loss of ERRa/g in hiPSC-CMs without changing cardiac specification as evidenced of GATA4 protein expression – GATA4 protein expression is not a good marker of cardiac specification. The percentage of cells that are cardiomyocytes (e.g. FACS for TNNT2) would be a better marker.

3. ERRa/g deficiency resulted in significant alterations in the ATAC-seq profile including regions

with both reduced (4510 peaks) and increased (2316 peaks) peak sizes (FDR>0.05, FC>|2|, Fig. 4c). – should be $|FC| > 2$.

4. Fig. 1f and S2, please provide evidence of effective ERRg overexpression by Ad-ERRg.

Reviewer #3 (Remarks to the Author):

The paper by Sakamoto et al, describes an interesting mechanism in which ERR γ works cooperatively with PGC-1 α and GATA4 to control expression of cardiac structural components, whereas ERR γ and PGC-1 α work cooperatively in the absence of GATA4 to influence mitochondrial biogenesis and metabolism. The study is well executed, involved many different assays/experiments and provides additional insight into the regulation of cardiac maturation. This work directly follows on from their 2020 Circulation Research paper (Sakamoto et al, PMID: 32212902) on the role of ERR γ in cardiac maturation, which does reduce the novelty of the mechanisms described within the manuscript. However, the GAT4 co-regulation and GATA4 mutant modelling are interesting additional insights. I only have a few concerns/comments.

Concerns:

1. The title of the manuscript is inaccurate, this paper is primarily concerned with the maturation of hiPSCs-derived cardiomyocytes and not the differentiation. It is stated within the paper that ERR KO does not affect cardiac specification nor does it change the fetal expression of cardiac genes (i.e. TNNI1, GATA4). It should be changed from differentiation to maturation.
2. The use of fatty acid based media (low glucose, high fatty acid concentration and no insulin) is commonly used to mature hiPSCs derived cardiomyocytes. Does culture within a fatty acid based media of ERR KO myocytes exacerbate or nullify maturation defects? qPCR of a few maturation markers would provide a useful comparison.
3. RPMI media used for culture has a low calcium concentration (0.4 mM) that is much lower than physiological levels and may alter maturation. How does this impact the study given that calcium/CAMK are critical transcriptional regulators in cardiomyocytes (PMID: 28963192)? qPCR of a few maturation markers would provide a useful comparison or this issue should be discussed.
4. Please state what the replicates are, i.e what the n number indicates? Number of biological replicates? Number of experiments?

RESPONSES TO REVIEWER COMMENTS

Reviewer #1

This is a well-written manuscript that describes exciting and important findings. The authors primarily overlaid and integrated large next generation sequencing data sets in order to uncover mechanisms whereby ERRgamma cooperates with GATA4 and PGC1-alpha to control cardiomyocyte structural and metabolic gene programs, respectively. The approaches, methods and data are, for the most part, explicitly explained and accessible. This manuscript is extensive and the data are convincing. The work seems to be ideally suited to and within the scope of Nature Communications. I have only minor suggestions for improvement.

We thank this reviewer for an insightful and constructive review. Itemized responses to the points raised are provided below:

1. The description of Fig. 1a in the text (Page 5) describes transcripts, whereas the data includes protein levels of ERRs. This should be changed to say ‘Transcripts encoding the ERR family of nuclear receptors as well as protein levels of ERRalpha and gamma...’

Thank you for identifying this inconsistency. We have changed the text to provide a more accurate denotation of transcripts and protein levels (page 5, paragraph 2).

2. Page 4 ‘Cistromic analyses in human induced pluripotent stem cell-derived cardiomyocytes (hiPSC-CMs) suggested that ERRg is a direct activator of adult contractile and ion channel genes in addition to regulating canonical target genes involved in mitochondrial energetic processes such as FAO and oxidative phosphorylation (OXPHOS).’ Please provide references for this statement.

We have added the following references for this statement as suggested (page 4, top paragraph).

- Scarpulla, R. C., Vega, R. B. & Kelly, D. P. Transcriptional integration of mitochondrial biogenesis. *Trends Endocrinol. Metab.* **23**, 459-466 (2012).
- Vega, R. B. & Kelly, D. P. Cardiac nuclear receptors: architects of mitochondrial structure and function. *J. Clin. Invest.* **127**, 1155-1164, (2017).
- Sakamoto, T. *et al.* A Critical Role for Estrogen-Related Receptor Signaling in Cardiac Maturation. *Circ. Res.* **126**, 1685-1702, (2020).

3. In the Discussion, there should be mention of the limitations of hiPSC-CMs as a model of chronic cardiac conditions such as heart failure. In particular, with regard to immature sarcomere structure, seeing as this is one of the key processes claimed to be regulated by

ERRg-PGC1a-GATA4 cooperation.

We have added statements regarding the limitations of hiPSC-CM in modeling the fetal re-programming of energy metabolic and sarcomeric proteins (page 21, top of page). In addition, please see the new data (requested by Reviewer 3) in which we have supplemented the media to further drive the maturity of the hiPSC-CMs (page 7, bottom paragraph-page 8; new Figures 1f-g). The results of these experiments are consistent with our original data and provide additional support for a key role of ERR signaling in cardiomyocyte maturation including in hiPSC-CMs.

4. Following on from the previous point, whilst reference is made to enrichment of binding motifs for GATA, MEF2 and ERRs in human embryonic hearts, the potential lack of translational relevance should also be noted. Evaluation of RNA-seq data from hearts of human CHD patients, such as in doi.org/10.1038/ncomms12824, would improve the manuscript.

Thank you for this suggestion. Access to the dataset generated in McKean et al. *Nat Commun* requires a complicated authorization. However, we utilized an RNA-seq dataset from human heart failure samples (heart failure with reduced ejection fraction or HFrEF) vs. normal function controls recently published in *Circulation* 2021 (Hahn et al. PMID: 33118835). We found that the gene expression levels of ERR α/γ and its coactivator PGC-1 α were significantly downregulated in the right ventricular HFrEF samples (page 19; new Supplementary Figure 8a). We intersected our RNA-seq dataset from ERR α/γ KO hiPSC-CMs with the HFrEF RNA-seq dataset and found that 24.9 % of downregulated genes in ERR α/γ KO hiPSC-CMs were also downregulated in HFrEF (page 19; new Supplementary Figure 8b). The corresponding enrichment analysis indicated that in HFrEF, the dysregulated ERR/PGC-1 α axis could contribute to deactivated mitochondrial oxidative metabolism (page 19; new Supplementary Figure 8c). Interestingly, levels of ERR target genes coding cardiac structural components (*TNNT2*, *TCAP*, *TMOD1*, *TTN*, and *MYL3*), ion transporters (*KCNJ8*, *KCNK3*, and *ATP1B1*), and Ca²⁺ handling protein (*ATP2A2*) were also downregulated in the HFrEF dataset, resulting in the enriched KEGG pathway term related to Cardiac Muscle Contraction (page 19; Supplementary Fig. 8c). Among this latter pathway, *TTN*, *KCNJ8*, *KCNK3*, *ATP1B1*, and *ATP2A2* were defined as shared ERR γ and GATA4 targets (new Supplementary Data 2). These findings together with the results of our hiPSC-CM studies further support the conclusion that the ERR/PGC-1 α axis in cooperation with GATA4 is essential for the transcriptional control of both mitochondrial oxidative metabolism and cardiac contractile function in normal and failing human adult hearts. We have also briefly addressed these findings in the revised Discussion section (page 24, last paragraph).

5. The sex of the source of the hiPSCs should be stated in the Methods section.

The hiPSCs used in this study were derived from a male. This information has been added in the Methods section under the “hiPSC culture system” sub-section (page 25, line 3).

6. For RNA-seq data, any minimal threshold reads/RPM applied in a minimal number of samples should be stated.

We have selected the genes that have at least 10 reads across all samples in one experiment for further analysis. We have added this information in the Methods section under the “RNA-seq analysis” sub-section (page 35, paragraph 2)

7. I believe a methods section on Gene Ontology analysis, including the programs used and statistical analysis and thresholds applied, is missing.

We apologize for this omission. We have added a description of the “Enrichment analysis” in Methods (page 40, paragraph 2).

8. The method used for multiple comparison testing to calculate FDR adjusted p-values in NGS data sets used should be stated (Bonferroni, Benjamini-Hochberg..?).

We have used the Benjamini-Hochberg FDR method as further emphasized in the Methods section (page 35, paragraph 2; page 35, bottom; page 36, paragraph 2; page 40, paragraph 1; page 40 paragraph 2).

9. As I understand it, Fig. 3C is reflecting data in Table S1. I think the clusters and/or z-score values should then be indicated in Table S1. A reference to which Suppl Tables data figures are included in figure legends would help orientate the reader.

Thank you for this suggestion. We have revised the legend for new Figure 3c and provided z-score of each gene in the heatmap as new Supplementary Table 1.

Reviewer #2

The authors previously reported that ERRa and ERRg, known to regulate mitochondrial gene expression in multiple tissues, also regulates postnatal maturation of cardiomyocytes, including expression of cardiomyocyte-specific genes in addition to mitochondrial genes. This raised the question of how these broadly deployed transcription factors could drive cardiomyocyte specific gene programs. The authors pursued this question in this study, using human iPSC-CMs as the primary model system. The authors found that ERRg interacts with GATA4, an important cardiac transcription factor. They demonstrate that ERRg and GATA4 are present in the same protein complexes, and that GATA4 potentiates ERRg transcriptional activity. ERRg-activated genes that were co-bound by GATA4 were enriched for mature cardiomyocyte specific genes, such as genes important for contraction, Ca²⁺ handling, and electrical activity, whereas those that were not co-bound by GATA4 were enriched for mitochondrial metabolism. Both gene classes were activated by the ERRg co-activator PGC1a.

Overall the main advances of this study are the demonstration that ERR activates cardiomyocyte specific genes and not just mitochondrial genes, and that it does through collaboration with GATA4. The general concept of general transcription factors adopting tissue specific activity through interaction with tissue-specific transcription factors is certainly not novel. Nevertheless this new information on ERR is interesting and valuable.

The manuscript had a number of weaknesses that decreased enthusiasm, as described below.

We thank this reviewer for an insightful and constructive review. We have conducted additional experiments and analyses to address this reviewer's questions and concerns. Itemized responses to the points raised are provided below:

Major points:

1. Data quality and analysis

- ***The GEO record shows that in most cases either duplicate or triplicate data were obtained. QC analyses of the data, such as correlation between replicates, was not presented.***

RNA-seq

We have checked the correlation between biological duplicate or triplicates in all RNA-seq datasets newly generated in this study. All samples in the same group were highly correlated ($r > 0.98$). We now refer to these quality control data in the Methods section under the "RNA-seq analysis" sub-section (page 35, paragraph 2).

ChIP-seq

We have checked the correlation between biological duplicates in H3K27ac ChIP-seq for WT and ERR1 KO hiPSC-CMs. The duplicate in each genotype is highly correlated (WT $r > 0.99$, ERR1 KO $r = 1.00$). In addition, we confirmed that the average H3K27ac ChIP-seq signals around transcriptional start sites of all genes has the expected shape in both WT and ERR1 KO hiPSC-CMs (new Supplementary Figure 3c). We have referred to these QC data in the Methods section under the "ChIP-seq analysis" sub-section (page 37, paragraph 1).

- ***Information about how replicates were combined to define consensus peaks and signals is absent.***

We have pooled the replicate data of each condition for final peak calling. We have added this information in the Methods under ChIP-seq analysis and ATAC-seq analysis sub-section (page 37, paragraph 1; page 40, paragraph 1).

- ***It would be helpful to provide more information about each dataset. For example, a volcano plot of RNA-seq data showing number of up and down regulated genes, and their definitions.***

We have added the volcano plots of RNA-seq data with ERR α /1 KO hiPSC-CMs (new Supplementary Figure 1c) and with ERR1 OE hiPSC-CMs (new Supplementary Figure 1g). We have also referred to the number of regulated genes with their definitions in both RNA-seq datasets (page 5, bottom; page 6, line 14).

- ***ChIP-seq and ATAC-seq: how many peaks were there? These numbers could be shown in Fig. S3, Fig. 3. The number of ATAC-seq peaks appears to be low, leading to questions about data quality. Does the data meet ENCODE guidelines for ATAC-seq data?***

We have added the peak number as requested (new Figure 3a and new Supplementary Figure 3a). Regarding ATAC-seq data quality, we have performed additional analyses. The number of called peaks largely depends on parameters of the peak caller algorithm. ENCODE ATAC-seq QC guidelines (<https://www.encodeproject.org/atac-seq/>) recommend at least 100,000 called peaks in individual replicates, and at least 50,000 final peaks for IDR-based QC procedure. We

sought a higher confidence peak set, so we adjusted the peak calling parameters and called 17,588 (Control), 17,117 (KO1), and 16,004 (KO6) peaks. We have performed the QC procedure according to ENCODE guidelines, including peak calling with less stringent thresholds. The comprehensive report is presented below. Our ATAC-seq datasets meet the high-quality standards set by ENCODE in the majority (6/8) of the key measures including signal level (Supplementary Figure 4a), fragment length distribution expected of ATAC signals (new Supplementary Figure 4b), and reproducibility (Supplementary Figure 4c).

sample name	aligned fragments	alignment rate	PBC1	NRF	Replicate peaks	IDR peaks	IDR: Self consistency ratio	IDR: Rescue ratio	FRIP
Cont1_1	48045764	94.04%	0.838884	0.452921	141468	38350	1.01998451	1.272725	0.04879
Cont1_2	65174874	94.70%	0.718493	0.40241					
Cont2_1	48435726	93.70%	0.840022	0.44916	139413				0.049178
Cont2_2	64491708	94.37%	0.724268	0.402348					
KO1-1_1	56510678	95.40%	0.887893	0.513959	109500	41276	1.07301239	1.161522	0.046308
KO1-1_2	75833868	96.05%	0.76399	0.457143					
KO1-2_1	50423950	94.78%	0.898968	0.536903	100957				0.044236
KO1-2_2	69848052	95.60%	0.779214	0.47989					
KO6-1_1	58790298	95.15%	0.890831	0.554557	105555	39577	1.19442941	1.145185	0.045915
KO6-1_2	80103988	95.93%	0.770554	0.493807					
KO6-2_1	41959014	94.94%	0.905038	0.603678	87959				0.046515
KO6-2_2	59218638	95.66%	0.790865	0.541079					

• *When discussing overlaps between region sets, the statistical significance of the overlap should be included (e.g. permutation test or hypergeometric test).*

We have added the statistical significance calculated by the hypergeometric test in the Homer package, of the overlapped regions between ChIP-seq peaks (new Figure 2a) or ChIP-seq and ATAC-seq peaks (new Figure 4a, b, and d). We have added the statistical method in the Methods section under Intersection analysis with ChIP-seq datasets (page 37, paragraph 2) and ATAC-seq analysis (page 40, paragraph 1).

• *For motif analysis and GO analysis, the appropriate background gene sets or regions should be used. For instance, if hiPSC-CM superenhancers have GO terms for cardiac muscle cell differentiation or myofibril assembly, is it surprising that the ERRg-GATA4 subset is also? What is the enrichment found in this subset when the remaining SEs are used as the background?*

GO enrichment analysis:

We have re-analyzed our RNA-seq datasets to detect significantly enriched GO or KEGG pathway terms using an appropriate gene background; we have used 8,893 genes fairly expressed in hiPSC-CMs (FPKM \geq 10) in our RNA-seq data. All enrichment analyses in this manuscript have been updated (new Figure 2d, 5d, 7g, Supplementary Figure 1d, 1h, 3b, 8c), and the

conclusions remain the same with this updated enrichment analyses. Accordingly, Methods section under “Enrichment analysis” sub-section has been updated (page 40, paragraph 2). To answer the Reviewer’s specific question related to ERR1 and GATA4 intersection, when using the remaining SEs for the enrichment analysis with ERR1 and GATA4 shared peaks (ERR1+GATA4), no terms were enriched given that nearly half of SEs (99/213 SEs) contain ERR1 and GATA4 peaks. With the new background genes expressed in hiPSC-CMs, the enrichment analysis with ERR1+GATA4 also demonstrated that ERR1 and GATA4 cooperate to drive ion transport and cardiac structural genes consistent with our original enrichment analysis (original Figure 5c).

Motif analysis:

The general motif enrichment analysis requires a set of background regions which is difficult to define in a tissue-specific manner. We have used the default background sequences generated by the Homer package, which are genomic DNA sequences that match the GC-content distribution of the input sequences. In some instances, it may be interesting to perform differential motif enrichment analysis between two well-defined sets of regions, as specifically suggested in the following questions regarding the enriched motifs for the intersected regions between ERR1 and H3K27ac regions or ERR1 and GATA4 shared regions. We agree and have conducted additional analyses, as outlined below. Accordingly, the Methods section under each intersection analysis sub-section has been updated (page 37, paragraph 1; page 37, bottom paragraph; page 38, top paragraph)

• Similar considerations for other GO and motif analyses. As another example, is there differential motif enrichment in regions with decreased H3K27ac in ERR KO, compared to regions with no change in H3K27ac? Specifically, is GATA4 motif enriched in all ERRg regions, or only in the ERRg subset with reduced H3K27ac?

To answer this question, we have now searched specifically for motifs enriched within ERR1 peaks exhibiting two different patterns of H3K27ac (by ERR1 KO): 1) ERR1 peaks with decreased H3K27ac by KO ERR1 (original Figure 2b and new Figure 2c) and 2) ERR1 peaks in which H3K27ac was not altered by KO ERR1 (Supplementary Figure 3d). The results indicate that GATA binding motif was enriched in ERR1 peaks regardless of H3K27ac deposition status (new Figure 2c and new Supplementary Figure 3d). It is possible that ERR1 is more important for H3K27Ac in a subset of enhancers.

2. Both ERR and GATA transcription factor families are redundant – ERR knockout requires ablation of both ERRa and ERRg, and GATA4 ablation requires ablation of both GATA4 and GATA6. This issue is not well addressed in this study, which focuses on ERRg-GATA4 interaction. Given the redundancy, one would expect that ERRg-GATA6, ERRa-GATA4, and ERRa-GATA6 interactions should also potentiate activation of the same sets of genes.

Thank you for this suggestion. We have performed new Gal4 DBD reporter experiments to address the potential interactions of ERRct/ERR1 with GATA4 and GATA6. The reporter studies demonstrated that ERRct cooperates with GATA4 and that this cooperativity is significantly greater than the ERR1/GATA4 interaction (page 16, bottom; new Supplementary Figure 7b). In

addition, the Gal4 DBD assays demonstrated interactions between GATA6 and ERR α or ERR γ , but GATA6 cooperativity with ERR α was significantly weaker than that of GATA4 (new Supplementary Figure 7b). These results indicate that under certain conditions ERR α or γ can interact with GATA6. We also analyzed a published GATA6 ChIP-seq dataset generated from hiPSC-CMs in the Seidman lab, as we did for GATA4 and ERR γ cooperativity. This analysis identified only a very small number (2.4 %) of GATA6 peaks (1,355/56,572 peaks) that overlapped ERR γ peaks and, conversely, only 4.1 % of ERR γ peaks (1,355/33,253 peaks) overlapped with GATA6 peaks (new Supplementary Figure 6e). None of pathway terms were enriched with ERR γ /GATA6-shared peaks, although a few cardiac genes including *MYL2*, *BIN1*, *KCNQ1*, *ATP2A2*, and *ACTA1* were annotated as targets (page 13, bottom and page 14, top paragraph; new Supplementary Fig. 6e). Taken together, we conclude that although GATA6 may be capable of interacting with ERR α or ERR γ under some experimental conditions (or perhaps in different cell types), the GATA4/ERR interaction would appear to be the major interaction in cardiomyocytes.

3. The manuscript demonstrates that over-expressed ERR γ and GATA4 co-precipitate. However, this should be demonstrated at the level of endogenous proteins, and appropriate controls to demonstrate IP specificity should be included. Evidence of direct ERR γ -GATA4 interaction is lacking.

Thank you for this suggestion. We have conducted new experiments to further delineate the ERR/GATA4/PGC-1 α interaction. Attempts at demonstrating interaction between the endogenous proteins have been unsuccessful, likely due to the lack of sensitivity of the ERR γ antibody. Accordingly, we conducted IP of overexpressed FLAG-tagged ERR γ in hiPSC-CMs and showed an interaction with endogenous GATA4 (page 17, bottom; new Figure 7d).

GATA4 was shown to potentiate transactivation by ERR γ in an assay that did not depend upon GATA4 binding activity. However, the motif enrichment and genome browser shots showing GATA4 and ERR γ binding to adjacent regions suggests that GATA4 DNA binding activity is also involved. Whether or not GATA4 and ERR γ influence each other's DNA binding activity and chromatin occupancy was not investigated. Indeed, "cobinding" in the current analysis is defined as both factors binding to the same gene, and whether there is significant co-binding to the same enhancers is not addressed.

We have conducted a series of experiments and analyses to further probe this interesting question as summarized below:

- Extended analysis of the GATA4 and ERR γ ChIP-seq dataset intersection. To further clarify ERR γ /GATA4 co-binding on the same genomic region, we re-analyzed the published GATA4 ChIP-seq and our ERR γ ChIP-seq datasets. In the original manuscript, we performed the intersection with annotated genes from each ChIP-seq dataset (original Figure 5b). In the revised manuscript, we also intersected each ChIP-seq dataset to define exact overlapped ERR γ and GATA4 peaks. 18.9 % of GATA4 peaks (4,460/23,598 peaks) overlapped with ERR γ specific peaks and 13.4 % of ERR γ peaks (4,460/33,253 peaks) overlapped with GATA4 peaks (page 12, middle; new Figure 5a). The enrichment analysis demonstrated that GATA4 and ERR γ co-bind on the DNA associated to cardiac

genes encoding contractile proteins, ion transporters, and Ca²⁺ handling proteins (page 13, paragraph 2; new Figure 5a and 5d) consistent with our original conclusion. The GATA4, ERR1, and SE intersection analysis was updated (page 13, paragraph 1; new Figure 5b). As also suggested, we examined the distance between ERR1 and GATA4 peaks. The analysis indicates that ERR1 and GATA4 peaks are colocalized within 200 bp in cardiac super-enhancer regions. The details of this analysis have been added to the revised manuscript (page 12, bottom; new Figure 5b and Supplementary Figure 6b)

- Transcriptional activity assays to further assess the requirement of DNA binding for ERR/GATA4 cooperativity. We have conducted additional experiments to further address the necessity of DNA binding for the ERR and GATA4 transcriptional regulatory cooperativity within the *TNNI3* regulatory region. Re-analysis of the *TNNI3* promoter region (defined by the ERR1/H3K27ac ChIP-seq analyses) using JASPAR (<http://jaspar.genereg.net/>) (Sandelin et al. PMID: 14681366) identified additional putative ERR response elements (ERREs) (schematic in new Figure 6f). Therefore, we repeated promoter-reporter experiments after mutating all of the potential ERREs in this region. The multi-ERRE site deletion *TNNI3* promoter mutant exhibited significantly decreased cooperative activation by GATA4 and ERR1 (page 15, main paragraph; new Figure 6g). In addition, as shown in the original manuscript, mutation of GATA4 binding sites in this region also reduced GATA4 and ERR1 cooperativity (original Figure 6d and new Figure 6g).

To complement the results of the *TNNI3* binding site mutants, we conducted new experiments with GATA4 DNA binding mutants. Notably, the C-terminal zinc finger of GATA4 is known to be essential for DNA binding (Morrissey et al. PMID: 9079680). Deletion of the C-terminal zinc finger region significantly attenuated ERR1 and GATA4 cooperativity on the *TNNI3* promoter reporter, whereas an N-terminal zinc finger mutant did not. These new results were added to the revised manuscript (page 15, main paragraph; new Figure 6h). In addition, the naturally-occurring G296S GATA4 mutant diminished ERR1 and GATA4 cooperation (page 18, main paragraph; new Figure 7f). The G296S mutation is located adjacent to the C-terminal zinc finger region and has been known to reduce GATA4 DNA binding capacity (Garg et al. PMID: 12845333). These collective data indicate that full ERR1 and GATA4 cooperativity requires each factor binding to the corresponding DNA binding sites, at least for the *TNNI3* regulatory region.

We have also now shown that a well-described enhancer region upstream of the *MYH6-7* cluster (Gacita et al. PMID: 33478249) is regulated by ERR1/GATA4/PGC-1 α . Overexpression of three factors cooperatively activated the activity of the *MYH6-7* enhancer using a reporter assay (page 15, bottom; new Figure 6i).

- Chromatin occupation assays to assess DNA binding cooperativity. As shown in the original manuscript we have conducted ChIP-qPCR experiments to assess the impact of GATA4 deficiency on ERR1 occupation on the *TNNI3* promoter region in hiPSC-CMs, and *vice versa*. Depletion of GATA4 significantly reduced ERR1 enrichment (original Supplementary Figure 5). Conversely, GATA4-HA enrichment was attenuated by ERR α /1 KO in hiPSC-CMs (original Figure 6b). In the revised manuscript, we have added the *MYH6-7* enhancer site as another example. Depletion of GATA4 in hiPSC-

CMs resulted in the significant reduction of ERR1 occupation on this enhancer site (page 14, paragraph 2, new Supplementary Figure 6g). GATA4-HA occupation tended to be decreased by KO ERR α /1 (page 15, paragraph 1, new Figure 6c), although the change was not significant. As shown for the *TNNI3* promoter and the *MYH6-7* enhancer, ERR1 occupation on this enhancer region was reduced with PGC-1 α depletion (page 17, paragraph 2, new Figure 7c). These data indicate that at least on the *TNNI3* and *MYH6-7* enhancer loci, GATA4 and ERR α /1 influence the DNA binding of the respective factor. These data are also consistent with the observed ERR1 and GATA4 physical interaction in the IP experiments (page 17, bottom; new Figure 7d).

Taken together, our original and new results indicate that the full cooperative interaction of ERR1 (or ERR α) with GATA4 requires: 1) DNA binding; 2) a direct interaction between ERR α /1 and GATA4; and 3) PGC-1 α coactivation of this complex via direct interaction with ERR α or ERR1 but not GATA4.

It is also notable that the two GATA4 mutants that significantly affected ERRg transcriptional activity both impair DNA binding. Could GATA4 DNA binding be influencing the activity of the GAL4 reporter?

Single GATA4 transfection did not affect activity of the Gal4 DBD itself or the ERR α /1-Gal4 reporter as indicated in new Figure 7a and new Supplementary Figure 7b.

Since SEs are large, co-binding of ERRg and GATA4 to the same SE does not provide information about potential cooperative binding. What is the distance between ERRg and GATA4 sites in SEs?

We have calculated the distance between closest ERR1 and GATA4 peaks in the same super-enhancers, as suggested. There were 99 super-enhancers containing both ERR1 and GATA4 peaks, and in 45%, GATA4 and ERR1 were co-localized within 200 base pairs (page 12, bottom; new Figure 5b), suggesting they often co-bind in the same super-enhancer regions. We have added the distribution of distances between closest ERR1 and GATA4 peaks in the same super-enhancers in new Supplementary Figure 6b.

4. “Enhancers” are usually considered distal regions, distinct from “Promoters”. However, this distinction is not made in this manuscript. For example, in Fig. 3b the selected examples are all promoters.

We have clarified this designation (page 9, paragraph 2; page 10, top; page 20, paragraph 2; page 21, paragraph 2). In the original manuscript, the *NPPA-NPPB* cluster region (Man et al. 33107387; original Figure 3b) was described as a distal enhancer example. We have also added one more cardiac distal enhancer example, the *MYH6-7* cluster region (Gacita et al. PMID: 33478249) (new Figure 3b).

5. The manuscript tends to jump into the analysis of specific examples without providing a more global view. To what extent do the conclusions made for these cherry-picked examples

hold in a more unbiased, genome-wide view?

In the original manuscript, we conducted unbiased analyses to determine the ERR γ targets such as *TNNI3* and *MYH7* using ERR γ /H3K27ac ChIP-seq, and ATAC-seq datasets. We have also integrated the published ChIP-seq datasets for GATA4 and MED1 occupations to determine ERR γ and GATA4 shared targets. To validate ERR γ and GATA4 cooperativity on cardiac genes, we focused on *TNNI3* given that *TNNI3* is a known robust cardiomyocyte maturation marker (Bedada et al. PMID: 25358788). In the revised manuscript, we have added *MYH6-7* enhancer as another example (new Figure 5e and 6i) for the validation study.

In addition to the original analyses, we have done additional unbiased analyses as suggested including motif distance between ERRE and GATA4 binding site (new Figure 5b; new Supplementary Figure 6b), and Cistrome DB toolkit analysis (new Figure 5c). All of these additional data also support our conclusion that ERR γ cooperates with GATA4 at cardiac enhancer regions.

6. The manuscript argues that ERR γ + GATA4 is important for the expression of maturation-related cardiomyocyte structural genes, with TNNI3 used as the example. The hiPSC-CM system is not ideal for studying maturation because maturation is arrested in this system. The authors have access to the relevant knockout mice. Can they provide data from the mouse system to demonstrate that ERR + GATA4 is required for maturation. Is there evidence that GATA4 is required for CM maturation?

This is an important point. Unfortunately, in vivo knockdown (KD) or overexpression of GATA4, an essential cardiac transcription factor, causes cardiac dysfunction in mice confounding conclusions regarding impact on maturation *per se*. For example, the Molkentin lab has reported that cardiac-specific deletion of GATA4 with β MHC or α MHC-Cre drivers results in a cardiomyopathy phenotype (Oka et al. PMID: 16514068). In addition, overexpression of GATA4 in mouse hearts with an α MHC promoter, causes significant cardiac hypertrophy and dysfunction (Liang et al. PMID: 11356841). Lastly, William Pu's group has reported that postnatal knockdown of GATA4 and GATA6 in mouse heart using AAV-Cre results in cardiomyopathy along with re-expression of the fetal gene program including upregulation of *Myh7* and downregulation of *Myh6* (Prendiville et al. PMID: 26023924). More recently, the Pu lab has reported that GATA4 and 6 KD in postnatal hearts with CRISPR/Cas9 AAV9 significantly increased activity of the fluorescence reporter driven by the *Myh7* [a fetal isoform of myosin in rodent hearts (VanDusen et al. PMID:34290256)] regulatory region. Taken together, these results suggest that GATA4 and GATA6 are indeed essential for the postnatal cardiac maturation program but given the heart failure phenotype (which also re-activates a fetal gene program) it is difficult to determine direct vs indirect effects of GATA4. Our recent studies of cardiac-specific ERR KD during the postnatal period in mice have been more definitive, demonstrating a significant primary impact on cardiac maturation (Sakamoto et al, PMID: 32212902).

Given the difficulties in studying the impact of GATA4 loss-of-function in vivo, we used a cell-based model (hiPSC-CMs) to directly assess GATA and ERR cooperation herein. To optimize the hiPSC-CM system, as suggested by Reviewer 3, we cultured wild-type and ERR α/γ KO hiPSC-CMs in a recently described "maturation cocktail" (Funakoshi et al. PMID: 34039977) to further drive maturation. We found that the "maturation cocktail", which includes

addition of palmitate and several hormones (triiodothyronine, GW4647: a PPAR α agonist, and dexamethasone), further induced cardiac maturation markers. ERR α/γ KO markedly suppressed the induced expression of the maturation markers in this optimized system (page 7, bottom paragraph-page 8; new Figure 1f and g). These results provide additional support for our conclusion that ERR α/γ is a critical driver of cardiomyocyte metabolic and structural maturation. However, to your point, even with this experimental modification, fully mature human cardiomyocytes are not achieved (Funakoshi et al. PMID: 34039977). This limitation is included in the Discussion (page 20, bottom-page 21, top paragraph).

7. What features distinguish ERRg peaks that are dependent or independent of GATA4? In addition to differential motif analysis, the authors could mine existing ChIP-seq data from hiPSC-CMs to define features or TF occupancy that differs between these ERRg peak classes.

As requested, we have performed differential motif analyses with sites enriched with both ERR γ and GATA4 peaks (ERR γ +GATA4) or ERR γ without GATA4 peaks (ERR γ -GATA4). In this new motif analysis, as suggested above we have used each region as a background to find specifically enriched motifs for either ERR γ +GATA4 or ERR γ -GATA4 peaks. In ERR γ +GATA4, binding motifs for several essential cardiac transcription factors including TEAD and TBX20 were enriched along with the GATA binding motif, while we did not find these motifs in ERR γ -GATA4 (page 12, middle; new Supplementary Figure 6a). In ERR γ -GATA4 sites, the ERR binding motif was highly enriched (new Supplementary Figure 6a) as expected.

Analyses with Cistrome Data Browser (Zheng et al. PMID: 30462313; Mei et al. PMID: 27789702) revealed that the ERR γ +GATA4 sites significantly overlapped with the published cistrome data sets for known SE markers (Hnisz et al. PMID: 24119843; Lovén et al. PMID: 23582323) including MED1, EP300, and BRD4 (page 12, bottom; new Figure 5c). ERR γ -GATA4 sites overlapped with general enhancer marks including EP300 and POLR2A (Heinz et al. PMID: 25650801) (new Figure 5c). GATA4 without ERR γ peaks (GATA4-ERR γ) overlapped with MED1 datasets as reported (Ang et al. PMID: 27984724) but not other enhancer marks (new Supplementary Figure 6c). These findings further support the preliminary conclusion that ERR γ + GATA4 often co-localize on cardiac SEs along with other transcription regulators including MEF2A, CEBPA, SIRT6, TAL1, STAT2, TBLXR1, SMARCA4, CDX2, MBD2, RUNX1, and ORC2 (new Figure 5c). Interestingly, ERR γ -GATA4 regions include binding sites for nuclear receptors including NR5A2, estrogen receptors, NR2C2, NR2F2, and RXR α/γ (new Figure 5c). Notably, MEF2 binding sites were also found on ERR γ /H3K27ac overlapped sites (new Figure 2c and Supplementary Figure 3d) and open chromatin sites regulated by ERR α/γ (new Figure 4h). MEF2 binding has been confirmed in mouse cardiac enhancers (Akerberg et al. PMID: 31659164; Hashimoto et al. PMID: 31080136). In addition, CEBPA has been implicated as a transcription factor to drive cardiomyocyte maturation (Uosaki et al. PMID: 26586429), suggesting that MEF2A and CEBPA could be potential co-regulators for ERR γ and GATA4 at cardiac enhancers. We have included much of these findings in the revised manuscript but have been careful about not over-concluding regarding this analysis given that future validation studies would be necessary.

7. Fig. 1.

a-b, normalization of all the data to the value at day 0 can lead to some confusion. For example, *ERRa* has no significant change during the time course, but it is expressed as robustly as *ERRg*. Can the data be shown as expression relative to house keeping gene at each time point without normalizing day 0 to 1.

We have normalized the levels of mRNA expression to *RPLP0* levels at each time point during the hiPSC-CM time-course (new Figure 1a and new Supplementary Figure 1a) as requested.

d, please indicate statistical significance.

We have defined statistical significance for the differentially expressed genes shown in the heatmap (new Figure 1c) in the corresponding legend.

8. Fig. 2.b. As controls, plot the H3K27ac density at non-ERRg peaks.

To present the control region for H3K27ac negative ERR1, we plotted H3K27ac deposition around GATA6 peaks (Sharma et al. PMID: 33054971), another cardiac transcription factor (page 8, paragraph 2; new Figure 2b) given that H3K27ac regions are large, diffuse, vary greatly in size and have no well-defined center of its ChIP-seq peaks. With this control plot, we have found that H3K27ac signals were higher around ERR1 than around GATA6 peaks, and that decreased H3K27ac signals by KO ERR1 occurred specifically around ERR1 peaks.

H3K27ac – ERRg overlapping regions can be classified into those associated with ERR-activated, ERR-suppressed, and ERR-unchanged genes. Motif analysis and H3K27ac plots could be made for these three classes.

As requested, we classified three different regions and performed additional motif analyses. We have plotted H3K27ac signals around ERR1 peaks based on the expression changes in ERR α /1 KO hiPSC-CMs as we have already presented in the original manuscript (original Figure 2f). The H3K27ac intensities were much higher around the ERR1 peaks associated with ERR-activated targets (downregulated genes in ERR α /1 KO hiPSC-CMs) compared to the ones around the ERR1 peaks close to either ERR-suppressed targets or all other regions (new Figure 2g). The corresponding motif analyses have been conducted with the appropriate background to define specifically enriched motifs in either ERR-activated or -suppressed targets; we used “ERR1 and H3K27ac-shared regions not associated with regulated genes in ERR α /1 KO hiPSC-CMs” as the background. As shown in new Supplementary Figure 3d, NR5A2, thyroid hormone receptor (THR), and MEF2 motif were enriched around ERR-activated targets (page 9, paragraph 1). In contrast, we did not see this trend around ERR-suppressed targets. Around ERR-suppressed targets, a very few motifs were weakly enriched such as IRF2 and RFX (Supplementary Figure 3e).

9. Fig. 4e. What accounts for the increased width of ATAC peaks in KO?

One plausible explanation is the higher occupancy of nucleosomes on the flanks of the peak region in wild-type control hiPSC-CMs, while their distribution is more uniformly spread around

ERR γ peak regions in ERR α/γ KO hiPSC-CMs. When the peak center is depleted of nucleosomes, they are often re-distributed to the flanks. Indeed, the ATAC signal within the 150 bp center of the peak was sharply decreased by ERR α/γ KO (new Figure 4f), suggesting that chromatin accessibility is reduced.

Please include other regions (e.g. H3K27ac+, ERRg-) as control regions to show the reduced ATAC signal is specific to ERRg regions.

We have included ATAC signal plots around GATA6 peaks (Sharma et al. PMID: 33054971) as the control regions for ATAC signal negative ERR γ peaks (page 10, bottom; new Figure 4f). The ATAC signals around GATA6 peaks did not exhibit the reduced ATAC signals in ERR α/γ KO hiPSC-CMs, and the overall level of DNA accessibility as determined by ATAC signal was lower there (page 10, bottom; new Figure 4f).

10. Fig. 7e. The two GATA4 point mutants that reduced activity of the reporter both are defective in DNA binding. I.

We are unable to address this point given the text was interrupted.

Minor

1. ERRg-deficiency impacted many fetal to adult isoform switches as demonstrated by marked reduction in expression of TNNI3 – only example provided is TNNI3. What are the other switches affected referred to by “many”.

We have revised the following sentence (page 6, paragraph 1). “ERR α/γ -deficiency impacted expression of a subset of well-established adult cardiac sarcomeric gene isoforms as demonstrated by marked reduction in expression of *TNNI3*, *MYH7*, and *MYL2* with only minimal impact on the fetal form, *TNNI1*”.

2. ...downregulated by loss of ERR α/g in hiPSC-CMs without changing cardiac specification as evidenced of GATA4 protein expression – GATA4 protein expression is not a good marker of cardiac specification. The percentage of cells that are cardiomyocytes (e.g. FACS for TNNT2) would be a better marker.

We agree that GATA4 is not the best indicator for cardiac specification. However, we have conducted FACS experiment with anti-TNNT2 in our previous publication (Sakamoto et al, PMID: 32212902) and confirmed that both WT and ERR α/γ KO hiPSC-CMs exhibited no differences in the number of TNNT2-positive cells. This data supports the conclusion that ERR α/γ KO does not change cardiac specification along with our current GATA4 and TNNI1 protein expression results. We have now referred to the results of the FACS experiment (page 6, paragraph 1) in the revised manuscript.

3. ERR α/g deficiency resulted in significant alterations in the ATAC-seq profile including regions with both reduced (4510 peaks) and increased (2316 peaks) peak sizes (FDR>0.05, FC>|2|, Fig. 4c). – should be |FC| > 2.

We have revised this as suggested (page 10, paragraph 2). Thank you for identifying this error.

4. Fig. 1f and S2, please provide evidence of effective ERR γ overexpression by Ad-ERR γ .

We have added representative immunoblot images to verify ERR γ overexpression by Ad-ERR γ (new Supplementary Figure 1f).

Reviewer #3

The paper by Sakamoto et al, describes an interesting mechanism in which ERR' works cooperatively with PGC-1 α and GATA4 to control expression of cardiac structural components, whereas ERR' and PGC-1 α work cooperatively in the absence of GATA4 to influence mitochondrial biogenesis and metabolism. The study is well executed, involved many different assays/experiments and provides additional insight into the regulation of cardiac maturation. This work directly follows on from their 2020 Circulation Research paper (Sakamoto et al, PMID: 32212902) on the role of ERR' in cardiac maturation, which does reduce the novelty of the mechanisms described within the manuscript. However, the GAT4 co-regulation and GATA4 mutant modelling are interesting additional insights. I only have a few concerns/comments.

We wish to thank this reviewer for the favorable and constructive critique. Our response to each of the items raised is provided below:

1. The title of the manuscript is inaccurate, this paper is primarily concerned with the maturation of hiPSCs-derived cardiomyocytes and not the differentiation. It is stated within the paper that ERR KO does not affect cardiac specification nor does it change the fetal expression of cardiac genes (i.e. TNNI1, GATA4). It should be changed from differentiation to maturation.

Thank you for the suggestion, we agree. The title was revised to include “maturation” instead of differentiation.

2. The use of fatty acid based media (low glucose, high fatty acid concentration and no insulin) is commonly used to mature hiPSCs derived cardiomyocytes. Does culture within a fatty acid based media of ERR KO myocytes exacerbate or nullify maturation defects? qPCR of a few maturation markers would provide a useful comparison.

We were also interested in manipulating the media to further drive hiPSC-CM maturation. As suggested, we have cultured our wild-type control and ERR α/γ KO hiPSC-CMs with media containing palmitate and several hormone ligands as described (Funakoshi et al. PMID: 34039977) for 7 days. Interestingly, this “maturation cocktail” significantly increased the levels of genes related to adult mitochondrial oxidative metabolism (*CPT1B*, *ACSL1*, *FABP3*, and *CKMT2*), cardiac structural components (*TNNI3*, *MYBPC3*, and *TTN*), and ion transport (*ATPIA3*), and Ca²⁺ handling function (*RYR2*). As predicted by the original data, the expression of these genes was significantly suppressed in ERR α/γ KO hiPSC-CMs (page 7, bottom paragraph-page 8; new Figure 1f). Reduced ACSL1 and TNNI3 protein expression in ERR α/γ KO hiPSC-CMs were also confirmed (new Figure 1g). Thank you for this suggestion.

3. RPMI media used for culture has a low calcium concentration (0.4 mM) that is much lower than physiological levels and may alter maturation. How does this impact the study given that calcium/CAMK are critical transcriptional regulators in cardiomyocytes (PMID: 28963192)? qPCR of a few maturation markers would provide a useful comparison or this issue should be discussed.

We used RPMI media for this study as it is widely employed for hiPSC-CM culture. The question raised is interesting although we believe it is beyond the scope of this study. Perhaps a future interesting experiment might be to assess whether transcription factors downstream of calcium signaling such as MEF2 cooperate with ERR to drive cardiac contractile maturation?

4. Please state what the replicates are, i.e what the n number indicates? Number of biological replicates? Number of experiments?

As noted in the Statistics section in Methods (page 41, top), the “n” denotes biological replicates.

REVIEWER COMMENTS

Reviewer #1 (Remarks to the Author):

The authors have nicely addressed my comments. Minor: there needs to be consistency with the use of PGC1 alpha (protein) vs. PPARGC1A (gene). For example, in Figure 7, RNA/gene are differentially referred to as PGC1 alpha and PPARGC1A in the same figure.

Reviewer #2 (Remarks to the Author):

Re: Sakamoto et al., "The Nuclear Receptor ERR Cooperates with the Cardiogenic Factor GATA4 to Orchestrate Cardiomyocyte Maturation"

The authors have revised the manuscript to address the reviewers' comments. The experiments are well done and make a valuable, although somewhat incremental, contribution.

Some remaining comments:

1. Lines 207-208: "In marked contrast to the ERR enhancer results, the vast majority of the ERRg-containing SEs located in proximal promoter regions were ...
 - SEs are several kb long, longer than promoters. This sentence needs to be rewritten.
2. For all of the overlaps reported, please give the p-value for the significance of the overlap. This was provided in some cases but not others. For example, Fig 3a, 5a, 7g, Suppl Fig 8b
3. Lines 209- 210 and Fig 3b: provide summary numbers for genes in each class
4. Fig 4h: Please provide p-values for motifs.
5. Line 257 and Suppl Fig 5c: Is the Smyd1 promoter used in the luc reporter bound by ERRg and contain an ERRg motif?
6. Cistrome Data Browser figures: these figures are difficult to evaluate. A number of TFs are shown which have Giggle scores comparable to or higher than some of the highlighted TFs, yet are not expressed in cardiomyocytes. For example, Gata1, Gata2, and Tal1. What is the specificity of the findings reported? Why does ERRg + GATA4 not have a high Giggle score for ESRRA, which is the top score for ERRg – GATA4? The authors highlight several TFs in the text, but it is not clear why these were selected. For example, in Fig 5c why are several factors colored blue? Whereas Tbx5 has a relatively high Giggle score but is not highlighted?
7. Fig 5d. Why are there no metabolic terms for ERRg – GATA4 regions?
8. For the superenhancer analysis, one wonders why the authors did not use H3K27ac to call superenhancers using the Rose algorithm, instead of Med1. They have H3K27ac signal in WT and ERRa/g KO, and therefore they would be able to directly evaluate the effect of ERR KO on superenhancers.
9. The transcriptomic comparison of human HF and ERRa/g KO iPSC-CMs is rather circumstantial. What is the evidence that the overlaps described reflect shared ERRg/GATA4/PGC1a regulation rather than coincidence?

Reviewer #3 (Remarks to the Author):

The revisions by Sakamoto et al, have addressed all of my concerns and is a thorough and insightful manuscript. This study provides further and much needed understanding of the processes governing cardiac maturation and disease, and is a welcome addition to the literature.

REVIEWER COMMENTS

Reviewer #1 (Remarks to the Author):

The authors have nicely addressed my comments. Minor: there needs to be consistency with the use of PGC1 alpha (protein) vs. PPARGC1A (gene). For example, in Figure 7, RNA/gene are differentially referred to as PGC1 alpha and PPARGC1A in the same figure.

We thank the Reviewer for this suggestion regarding labeling consistency. The corresponding Results (page 17, paragraph 2) and Methods sections (page 28, paragraph 2) have been revised as suggested.

Reviewer #2 (Remarks to the Author):

The authors have revised the manuscript to address the reviewers' comments. The experiments are well done and make a valuable, although somewhat incremental, contribution.

We thank this Reviewer for the additional helpful input and comments. Our itemized response to each issue raised is provided below.

Some remaining comments:

1. Lines 207-208: “In marked contrast to the ERR enhancer results, the vast majority of the ERRg-containing SEs located in proximal promoter regions were ...”

•SEs are several kb long, longer than promoters. This sentence needs to be rewritten.

We agree that this sentence is inaccurate as written. The sentence has been revised as indicated here; “the vast majority of the ERR□-containing SEs overlapped with proximal promoter regions linked to...” (page 9, paragraph 2, middle part).

2. ***For all of the overlaps reported, please give the p-value for the significance of the overlap. This was provided in some cases but not others. For example, Fig 3a, 5a, 7g, Suppl Fig 8b.***

As requested, we have added *p*-values for the overlaps in each Figure (New Figures 3a, 5a, 7g; Supplemental Figures 6e and 8b). We have updated the corresponding Methods section accordingly (page 36, paragraph 2).

3. ***Lines 209- 210 and Fig 3b: provide summary numbers for genes in each class.***

We have added the number of genes for the cardiac structural protein and metabolic groups (twenty cardiac-structural related genes and only four metabolic genes; page 9, paragraph 2).

4. ***Fig 4h: Please provide p-values for motifs.***

We have added *p*-values for the motif analysis in the ATAC-seq experiment (New Figure 4h).

5. ***Line 257 and Suppl Fig 5c: Is the Smyd1 promoter used in the luc reporter bound by ERR α and contain an ERR α motif?***

As explained in the legend for Supplemental Figure 5c, we have cloned the *SMYD1* promoter region that contained regions occupied by ERR α based on our ChIP-seq analyses. In addition, we have scanned the promoter region using JASPAR (<http://jaspar.genereg.net/>) (Sandelin et al. PMID: 14681366) and identified many predicted ERR binding motifs in this cloned region as shown below. Accordingly, we have revised the corresponding Results section (page 11, bottom paragraph) and have added the genomic position (chr2:88066119-88069666) for this promoter region to the corresponding Figure legend to serve as a reference.

Name	Score	Strand	Predicted sequence
MA0643.1.Esrrg	12.0307	-	ccaaggtcgt
MA0592.3.ESRRA	10.5543	-	ggccaaggtcgtg
MA0141.3.ESRRB	10.0109	-	ccaaggtcgtg
MA0643.1.Esrrg	8.86764	-	acaaggtgat
MA0141.3.ESRRB	8.83511	+	ccatggtcatt
MA0592.3.ESRRA	8.64499	+	acctatggtcatt
MA0592.3.ESRRA	8.18907	-	aaacaaggtgata
MA0592.3.ESRRA	8.06489	+	tctgaaggccacc
MA0592.3.ESRRA	7.94538	+	ggtgaaggtgagc
MA0141.3.ESRRB	7.63058	-	acaaggtgata
MA0643.1.Esrrg	7.56303	+	tgaaggtgag
MA0592.3.ESRRA	7.28348	+	tctccagggcaca
MA0643.1.Esrrg	6.93281	+	tgaaggccac

MA0141.3.ESRRB	6.64181	+	tgaaggccacc
MA0643.1.Esrrg	6.61903	+	ccatggcat
MA0643.1.Esrrg	6.21123	+	agaaggacag
MA0643.1.Esrrg	5.22332	-	aaaatgcat
MA0643.1.Esrrg	4.07538	-	ctgaggcag
MA0643.1.Esrrg	4.04448	+	ccaaggtggg
MA0643.1.Esrrg	3.65711	+	tggaggctt
MA0643.1.Esrrg	3.41341	-	gcaaggatag
MA0643.1.Esrrg	3.39068	-	tcaaggaaa
MA0643.1.Esrrg	3.26134	+	gcaatgaaa
MA0643.1.Esrrg	3.13314	-	acaaggtga

6. Cistrome Data Browser figures: these figures are difficult to evaluate. A number of TFs are shown which have Giggie scores comparable to or higher than some of the highlighted TFs, yet are not expressed in cardiomyocytes. For example, *Gata1*, *Gata2*, and *Tal1*. What is the specificity of the findings reported?

The Cistrome Data Browser (DB) Tool kit uses cistrome datasets collected from various tissues/cell types to define the similarity between genomic datasets. Thus, Cistrome DB can introduce some factors with low expression in the tissue of interest. We have revised the figures with the expression criteria to remove non-cardiac enriched transcription factors. Specifically, we have confirmed that *GATA1*, *NR5A2*, *CDX2*, and *KRAB* levels are deficient in hiPSC-CMs, and, therefore, have been removed from the Figures. The updated Figures now list transcriptional regulators that are expressed in hiPSC-CMs (FPKM>0.1) based on our RNA-seq data from wild-type control hiPSC-CMs (New Figure 5c and Supplemental Figure 6c). The main text (page 13, paragraph 1, line 8) and Methods section (page 39, paragraph 1) have been revised accordingly. Thank you for pointing this out.

Why does *ERRg + GATA4* not have a high Giggie score for *ESRRA*, which is the top score for *ERRg - GATA4*?

The *ESRRA*'s GIGGLE score reflects the correlation of the entire set of *ESRRA* peaks from CistromeDB with a given region set. Because the *ERR1+GATA4* regions represent a much smaller, specific subset of all *ERR1* peaks, several factors that bind these regions received relatively higher scores than for *ESRRA*. Therefore, *ESRRA* was found lower on the *ERR1+GATA4* rank list and was not included in Figure 5c. Specifically, the GIGGLE score of *ESRRA* in *ERR1+GATA4* is 973.4, which is slightly lower with the score (1093.8) in *ERR1-GATA4*, yet the enrichment of *ESRRA* in *ERR1+GATA4* is still significant ($p=4.2157E-149$). We now have revised the Figure 5c to include *ESRRA* in *ERR1+GATA4* and updated the corresponding Results section (page 12, bottom paragraph - page 13; new Figure 5c).

The authors highlight several TFs in the text, but it is not clear why these were selected. For example, in Fig 5c why are several factors colored blue? Whereas Tbx5 has a relatively high Gigggle score but is not highlighted?

We agree that the color coding, which was meant to highlight group-specific factors, could lead to some confusion. TBX5 is listed in both ERR1+GATA4 (Figure 5c) and GATA4-ERR1 (Supplemental Figure 6c) and was not highlighted. We have revised the graphs so that they now display in a simpler format without coloring (new Figure 5c and new Supplemental Figure 6c), and the Figure legends have been updated. In addition, the main text highlights specific overlapping transcription factors in the ERR1+GATA4 dataset (page 13, paragraph 1). We thank the Reviewer for this point.

7. Fig 5d. Why are there no metabolic terms for ERRg – GATA4 regions?

We apologize for an error in the Figure 5d legend that caused this confusion. Indeed, the bar graphs in Figure 5d represent the enrichment score for pathways with ERR1+GATA4 regions rather than ERR1-GATA4 as denoted in the last version. We have now corrected the Figure 5d legend. Thank you for identifying this discrepancy.

8. For the superenhancer analysis, one wonders why the authors did not use H3K27ac to call superenhancers using the Rose algorithm, instead of Med1. They have H3K27ac signal in WT and ERRa/g KO, and therefore they would be able to directly evaluate the effect of ERR KO on superenhancers.

The Srivastava lab had already determined cardiac super-enhancer (SE) regions based on Mediator Complex Subunit 1 (MED1) ChIP-seq in hiPSC-CMs (Ang et al. PMID: 27984724). In addition, the Richard Young lab has also reported that MED1 occupation performs optimally to distinguish general enhancers and SEs (Whyte et al. PMID: 23582322), although other enhancer features such as H3K27ac deposition has also been used as pointed out by the Reviewer. We utilized the published SEs by Dr. Srivastava lab for our analysis. It would be of interest to determine the effects of ERR KO on H3K27Ac deposition in hiPSC-CM SEs across the genome as suggested. However, this analysis would likely add a significant amount of additional data to this manuscript and we believe that our conclusions would not be changed by this analysis. We are interested in conducting future experiments that combine ERR KO with selective loss-of-function of other relevant cardiogenic TFs (e.g. GATA4, MEF2) during developmental maturation and/or in disease states. However, we believe such studies are beyond the scope of this manuscript.

9. The transcriptomic comparison of human HF and ERRa/g KO iPSC-CMs is rather circumstantial. What is the evidence that the overlaps described reflect shared ERRg/GATA4/PGC1a regulation rather than coincidence?

We thank the Reviewer for this query. The following lines of evidence support our conclusion: 1) we have found that ERRa, ERR1, and PGC-1a expression levels are significantly downregulated in HFrEF hearts (page 19, paragraph 2, Supplemental Figure 8a); 2) we have shown that ERR1 and GATA4 functionally cooperate and physically interact as demonstrated by

reduced GATA4 DNA binding on a subset of cardiac-enriched targets with depletion of ERRa/y (page 15, paragraph 1; Figure 6c), further corroborated by luciferase reporter and immunoprecipitation studies (page 15, paragraph 2-page 16, paragraph 1; Figure 6d-i and page 17, bottom paragraph-page 18, paragraph 1; Figure 7d and Supplemental Figure 7c). In addition, depletion of PGC-1a led to decreased ERRy recruitments on similar cardiac gene sets (page 17, paragraph 2; Figure 7c); and 3) the expression of ERR downstream targets was found to be reduced in HFrEF as supported by our intersection analysis and followed enrichment analysis with the two RNA-seq datasets (page 19, paragraph 2; Supplemental Figure 8c).

To provide further evidence, we have now conducted additional statistical analyses (as also suggested in this Reviewer's point #2) on the intersection analysis to more rigorously and objectively define the commonly downregulated genes between ERRa/y KO hiPSC-CM and the HFrEF datasets. We found that the downregulated genes in each RNA-seq dataset overlapped significantly ($p=1.85e-84$) rather than coincidentally, further supporting the conclusion that ERRa/y-mediated transcriptional regulation is altered in HFrEF (new Supplemental Figure 8b). In addition, we found that several genes encoding cardiac structural components, ion transporters, and Ca²⁺ handling protein were also downregulated in ERRa/y KO hiPSC-CMs and HFrEF subset, and a subset (*TTN*, *KCNJ8*, *KCNK3*, *ATP1B1*, and *ATP2A2*) were defined as ERRy+GATA4 targets (page 19, bottom paragraph; Supplemental Data 2).

As with all such analyses using human tissue data, these results will require future replication using independent datasets in order to arrive at a definitive conclusion. Accordingly, we have now included this caveat in the revised Discussion (page 25, paragraph 1).

Reviewer #3 (Remarks to the Author):

The revisions by Sakamoto et al, have addressed all of my concerns and is a thorough and insightful manuscript. This study provides further and much needed understanding of the processes governing cardiac maturation and disease, and is a welcome addition to the literature.

We thank this Reviewer for critical and thoughtful input.

REVIEWERS' COMMENTS

Reviewer #2 (Remarks to the Author):

The authors have responded to the prior comments. I have no further comments and I believe the manuscript is suitable for publication in Nature Communications.

REVIEWERS' COMMENTS

Reviewer #2 (Remarks to the Author):

The authors have responded to the prior comments. I have no further comments and I believe the manuscript is suitable for publication in Nature Communications.

We thank this reviewer for input that has strengthened this manuscript.